# Towards Robust Neural Networks via Close-loop Control

**Zhuotong Chen**[1][§]**, Qianxiao Li**[2,3][§]**, Zheng Zhang**[1]

[1]Department of Electrical & Computer Engineering, University of California, Santa Barbara, CA 93106
[2]Department of Mathematics, National University of Singapore, Singapore
[3]Institute of High Performance Computing, A*STAR, Singapore
`ztchen@ucsb.edu, qianxiao@nus.edu.sg, zhengzhang@ece.ucsb.edu`

## Abstract

Despite their success in massive engineering applications, deep neural networks are vulnerable to various perturbations due to their black-box nature. Recent study has shown that a deep neural network can misclassify the data even if the input data is perturbed by an imperceptible amount. In this paper, we address the robustness issue of neural networks by a novel close-loop control method from the perspective of dynamic systems. Instead of modifying the parameters in a fixed neural network architecture, a close-loop control process is added to generate control signals adaptively for the perturbed or corrupted data. We connect the robustness of neural networks with optimal control using the geometrical information of underlying data to design the control objective. The detailed analysis shows how the embedding manifolds of state trajectory affect error estimation of the proposed method. Our approach can simultaneously maintain the performance on clean data and improve the robustness against many types of data perturbations. It can also further improve the performance of robustly trained neural networks against different perturbations. To the best of our knowledge, this is the first work that improves the robustness of neural networks with close-loop control [1].

## 1 Introduction

Due to the increasing data and computing power, deep neural networks have achieved state-of-the-art performance in many applications such as computer vision, natural language processing and recommendation systems. However, many deep neural networks are vulnerable to various malicious perturbations due to their black-box nature: a small (even imperceptible) perturbation of input data may lead to completely wrong predictions (Szegedy et al., 2013; Nguyen et al., 2015). This has been a major concern in some safety-critical applications such as autonomous driving (Grigorescu et al., 2020) and medical image analysis (Lundervold & Lundervold, 2019). Various perturbations have been reported, including the $\ell_p$ norm based attack (Madry et al., 2017; Moosavi-Dezfooli et al., 2016; Carlini & Wagner, 2017), semantic perturbation (Engstrom et al., 2017) etc. On the other side, some algorithms to improve the robustness against those perturbations have shown great success (Madry et al., 2017). However, most robustly trained models are tailored for certain types of perturbations, and they do not work well for other types of perturbations. Khoury & Hadfield-Menell (2018) showed the non-existence of optimal decision boundary for any $\ell_p$-norm perturbation.

Recent works (E, 2017; Haber & Ruthotto, 2017) have shown the connection between dynamical systems and neural networks. This dynamic system perspective provides some interesting theoretical insights about the robustness issue. Given a set of data $\mathbf{x}_0 \in \mathbb{R}^d$ and its labels $\mathbf{y} \in \mathbb{R}^l$ with a joint distribution $\mathcal{D}$, training a neural network can be considered as following

$$\min_{\boldsymbol{\theta}} \mathbb{E}_{(\mathbf{x}_0, \mathbf{y}) \sim \mathcal{D}} [\Phi(\mathbf{x}_T, \mathbf{y})], \quad \text{s.t. } \mathbf{x}_{t+1} = f(\mathbf{x}_t, \boldsymbol{\theta}_t),$$

---

[§]Equal contributing authors.
[1]A Pytorch implementation can be found in:`https://github.com/zhuotongchen/Towards-Robust-Neural-Networks-via-Close-loop-Control.git`

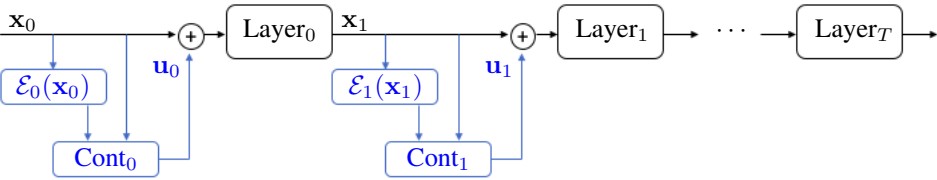

Figure 1: The structures of feed-forward neural network (black) and the proposed method (blue).

where $\boldsymbol{\theta}$ are the unknown parameters to train, and $f$, $\Phi$ represent the forward propagation rule and loss function (e.g. cross-entropy) respectively. The dynamical system perspective interprets the vulnerability of neural networks as a system instability issue, which addresses the state trajectory variation under small perturbations applied on initial conditions. The optimal control theory focuses on developing a control model to adjust the system state trajectory in an optimal manner. The first work that links and extends the classical back-propagation algorithm using optimal control theory was presented in Li et al. (2017), where the direct relationship between the Pontryagin's Maximum Principle (Kirk, 1970) and the gradient based network training was established. Ye et al. (2019) used control theory to adjust the hyperparameters in the adversarial training algorithm. Han et al. (2018) established the mathematical basis of the optimal control viewpoint of deep learning. These existing works on algorithm development are *open-loop control methods* since they commonly treat the network weights $\boldsymbol{\theta}$ as control parameters and keep them fixed once the training is done. The fixed control parameters $\boldsymbol{\theta}$ operate optimally for data sampled from the data distribution $\mathcal{D}$. However, various perturbation methods cause data distributions to deviate from the true distribution $\mathcal{D}$ (Song et al., 2017) and cause poor performance with the fixed open-loop control parameters.

## 1.1 PAPER CONTRIBUTIONS

To address the limitation of using *open-loop control methods*, we propose the **C**lose- **L**oop **C**ontrol **N**eural **N**etwork (**CLC-NN**), the first close-loop control method to improve the robustness of neural networks. As shown in Fig. 1, our method adds additional blocks to a given $T$-layer neural network: embedding functions $\mathcal{E}_t$, which induce running losses in all layers that measure the discrepancies between true features and observed features under input perturbation, then control processes generate control variables $\mathbf{u}_t$ to minimize the total running loss under various data perturbations. The original neural network can be designed by either standard training or robust training. In the latter case, our CLC-NN framework can achieve extra robustness against different perturbations. The forward propagation rule is thus modified with an extra control parameter $\mathbf{u}_t \in \mathbb{R}^{d'}$

$$\mathbf{x}_{t+1} = f(\mathbf{x}_t, \boldsymbol{\theta}_t, \mathbf{u}_t).$$

Fig. 1 should not be misunderstood as an open-loop control. From the perspective of dynamic systems, $\mathbf{x}_0$ is an initial condition, and the excitation input signal is $\mathbf{u}_t$ (which is 0 in a standard feed-forward network). Therefore, the forward signal path is from $\mathbf{u}_t$ to the internal states $\mathbf{x}_t$ and then to the output label $\mathbf{y}$. The path from $\mathbf{x}_t$ to the embedding function $\mathcal{E}_t(\mathbf{x}_t)$ and then to the excitation signal $\mathbf{u}_t$ forms a feedback and closes the whole loop.

The technical contributions of this paper are summarized below:

- The proposed method relies on the well accepted assumption that the data and hidden state manifolds are low dimensional compared to the ambient dimension (Fefferman et al., 2016). We study the geometrical information of the data and hidden layers to define the objective function for control. Given a trained $T$-layer neural network, a set of embedding functions $\mathcal{E}_t$ are trained off-line by minimizing the reconstruction loss $\|\mathcal{E}(\mathbf{x}_t) - \mathbf{x}_t\|$ over some clean data from $\mathcal{D}$ only. The embedding functions support defining a running loss required in our control method.

- We define the control problem by dynamic programming and implement the online iterative solver based on the Pontryagin's Maximum Principle to avoid the curse of dimensionality. The proposed close-loop control formulation does not require prior information of the perturbation.

- We provide a theoretical error bound of the controlled system for the simplified case with linear activation functions and linear embedding. This error bound reveals how the close-loop control improves neural network robustness in the simplest setting.

## 2 RELATED WORKS

Many techniques have been reported to improve the robustness of neural networks, such as data augmentation (Shorten & Khoshgoftaar, 2019), gradient masking (Liu et al., 2018), etc. We review adversarial training and reactive defense which are most relevant to this work.

**Adversarial Training.** Adversarial training is (possibly) the most popular robust training method, and it solves a min-max robust optimization problem to minimize the worse-case loss with perturbed data. Adversarial training effectively regularizes the network's local Lipschitz constants of the loss surface around the data manifold (Liu et al., 2018). Zhang et al. (2019) formulated the robustness training using the Pontryagon's Maximum Principle, such *open-loop control methods* result in a set of fixed parameters that operates optimally on the considered perturbation. Liu et al. (2020a;b) considered a close-loop formulation from the differential dynamic programming perspective, this algorithm is categorized as a *open-loop control method* because it utilizes the state feedback information to boost the training convergence and results in a set of fixed controls for any unseen data. On the contrary, the proposed CLC-NN formulation adaptively targets on the inputs with different control parameters and is capable of distinguishing clean data by generating no control.

**Reactive Defense.** A reactive defense method tries to reject or pre-process the input data that may cause mis-classifications. Metzen et al. (2017) rejected perturbed data by using adversarial detectors that are trained with adversarial data to detect abnormal data during forward propagation. Song et al. (2017) estimated the input data distribution $\mathcal{D}$ with a generative model (Oord et al., 2016) to detect data that does not belong to $\mathcal{D}$, it applies a greedy method to search the local neighbour of input data for a more statistically plausible counterpart. This purification process has shown improved accuracy with adversarial data contaminated by various types of perturbations. Purification can be considered as a one-step method to solve the optimal control problem that has the objective function defined over the initial condition only. On the contrary, the proposed CLC-NN solves the control problem by the dynamic programming principle and its objective function is defined over the entire state trajectory, which guarantees the optimality for the resulted controls.

## 3 THE CLOSE-LOOP CONTROL FRAMEWORK FOR NEURAL NETWORKS

Now we present a close-loop optimal control formulation to address the robustness issue of deep learning. Consider a neural network consisting of model parameters $\boldsymbol{\theta}$ equipped with external control policy $\boldsymbol{\pi}$, where $\boldsymbol{\pi} \in \boldsymbol{\Pi}$ is a collection of functions $\mathbb{R}^d \to \mathbb{R}^{d'}$ acting on the state and outputting the control signal. The feed-forward propagation in a $T$-layer neural network can be represented as

$$\mathbf{x}_{t+1} = f(\mathbf{x}_t, \boldsymbol{\theta}_t, \boldsymbol{\pi}_t(\mathbf{x}_t)), \quad t = 0, \cdots, T-1. \tag{1}$$

Given a trained network, we solve the following optimization problem

$$\min_{\overline{\boldsymbol{\pi}}} \mathbb{E}_{(\mathbf{x}_0, \mathbf{y}) \sim \mathcal{D}} \left[ J(\mathbf{x}_0, \mathbf{y}, \overline{\boldsymbol{\pi}}) \right] := \min_{\overline{\boldsymbol{\pi}}} \mathbb{E}_{(\mathbf{x}_0, \mathbf{y}) \sim \mathcal{D}} \left[ \Phi(\mathbf{x}_T, \mathbf{y}) + \sum_{s=0}^{T-1} \mathcal{L}(\mathbf{x}_s, \boldsymbol{\pi}_s(\mathbf{x}_s)) \right], \quad s.t. \text{ Eq. (1)}, \tag{2}$$

where $\overline{\boldsymbol{\pi}}$ collects the control policies $\boldsymbol{\pi}_0, \cdots, \boldsymbol{\pi}_{T-1}$ for all layers. Note that (2) differs from the open-loop control used in standard training. An open-loop control that treats the network parameters as control variables seeks for a set of *fixed* parameters $\boldsymbol{\theta}$ to match the output with true label $\mathbf{y}$ by minimizing the terminal loss $\Phi$, and the running loss $\mathcal{L}$ defines a regularization for $\boldsymbol{\theta}$. However, the terminal and running losses play different roles when our goal is to improve the robustness of a neural network by generating some adaptive controls for different inputs.

**Challenge of Close-loop Control for Neural Networks.** Optimal control has been well studied in the control community for trajectory optimization, where one defines the running loss as the error between the actual state $\mathbf{x}_t$ and a reference state $\mathbf{x}_{t,\text{ref}}$ over time interval $[0, T]$. The resulting control policy adjusts $\mathbf{x}_t$ and makes it approach $\mathbf{x}_{t,\text{ref}}$. In this paper, we apply the idea of trajectory optimization to improve the robustness of a neural network via adjusting the undesired state of $\mathbf{x}_t$. However, the formulation is more challenging in neural networks: we do not have a "reference" state during the inference process, therefore it is unclear how to define the running loss $\mathcal{L}$.

In the following, we investigate manifold embedding of the state trajectory to precisely define the loss functions $\Phi$ and $\mathcal{L}$ of Eq. (2) required for the control objective function of a neural network.

### 3.1 Manifold Learning for State Trajectories

**State Manifold.** Our controller design is based on the "manifold hypothesis": real-world high dimensional data can often be embedded in a lower dimensional manifold $\mathcal{M}$ (Fefferman et al., 2016). Indeed, neural networks extract the embedded features from $\mathcal{M}$. To fool a well-trained neural network, the perturbed data often stays away from the data manifold $\mathcal{M}$ (Khoury & Hadfield-Menell, 2018). We consider the data space $Z$ ($\mathbf{x} \in Z, \forall \mathbf{x} \sim \mathcal{D}$) as: $Z = Z_{\parallel} \bigoplus Z_{\perp}$, where $Z_{\parallel}$ contains the embedded manifold $\mathcal{M}$ and $Z_{\perp}$ is the orthogonal complement of $Z_{\parallel}$. During forward propagation, the state manifold embedded in $Z_{\parallel}$ varies at different layers due to both the nonlinear activation function $f$ and state dimensionality variation. Therefore, we denote $Z^t = Z_{\parallel}^t \bigoplus Z_{\perp}^t$ as the state space decomposition at layer $t$ and $\mathcal{M}_t \in Z_{\parallel}^t$. Once an input data is perturbed, the main effects of causing misclassifications are in $Z_{\perp}$. Therefore, it is important to measure how far the possibly perturbed state $\mathbf{x}_t$ deviates from the state manifold $\mathcal{M}_t$.

**Embedding Function.** Given an embedding function $\mathcal{E}_t$ that encodes $\mathbf{x}_t$ onto the lower-dimensional manifold $\mathcal{M}_t$ and decodes the result back to the full state space $Z_t$, the reconstruction loss $\|\mathcal{E}_t(\mathbf{x}_t) - \mathbf{x}_t\|$ measures the deviation of the possibly perturbed state $\mathbf{x}_t$ from the manifold $\mathcal{M}_t$. The reconstruction loss is nonzero as long as $\mathbf{x}_t$ has components in $Z_{\perp}^t$. The embedding functions are constructed offline by minimizing the total reconstruction losses over a clean training data set.

- **Linear Case:** $\mathcal{E}_t(\cdot)$ can be considered as $\mathbf{V}_t^r (\mathbf{V}_t^r)^T$ where $\mathbf{V}_t^r$ forms an orthonormal basis for $Z_{\parallel}^t$. Specifically one can first perform a principle component analysis over a collection of hidden states at layer $t$, then $\mathbf{V}_t^r$ can be obtained as the first $r$ columns of the resulting eigenvectors.

- **Nonlinear Case:** we choose a convolutional auto-encoder (detailed in Appendix B) to obtain a representative manifold embedding function $\mathcal{E}_t$ due to its ease of implementation. Based on the assumption that most perturbations are in the $Z_{\perp}$ subspace, the embeddings are effective to detect the perturbations as long as the target manifold is of a low dimension. Alternative manifold learning methods such as Izenman (2012) may also be employed.

### 3.2 Formulation for the Close-Loop Control of Neural Networks

**Control Objectives.** The above embedding function allows us to define a **running loss** $\mathcal{L}$:

$$\mathcal{L}(\mathbf{x}_t, \boldsymbol{\pi}_t(\mathbf{x}_t), \mathcal{E}_t(\cdot)) = \|\mathcal{E}_t(\mathbf{x}_t) - \mathbf{x}_t\|_2^2 + (\boldsymbol{\pi}_t(\mathbf{x}_t))^T \mathbf{R} \boldsymbol{\pi}_t(\mathbf{x}_t). \tag{3}$$

Here the matrix $\mathbf{R}$ defines a regularization term promoting controls of small magnitudes. In practical implementations, using a diagonal matrix $\mathbf{R}$ with small elements often helps to improve the performance. Now we are ready to design the control objective function of CLC-NN. Different from a standard open-loop control, this work sets the terminal loss $\Phi$ as zero because no true label is given during inference. Consequently, the **close-loop control** formulation in Eq. (2) becomes

$$\min_{\overline{\boldsymbol{\pi}}} \mathbb{E}_{(\mathbf{x}_0, \mathbf{y}) \sim \mathcal{D}} \left[ J(\mathbf{x}_0, \mathbf{y}, \overline{\boldsymbol{\pi}}) \right] := \min_{\overline{\boldsymbol{\pi}}} \mathbb{E}_{(\mathbf{x}_0, \mathbf{y}) \sim \mathcal{D}} \sum_{t=0}^{T-1} \left[ \mathcal{L}(\mathbf{x}_t, \boldsymbol{\pi}_t(\mathbf{x}_t), \mathcal{E}_t(\cdot)) \right], \quad s.t. \ Eq. \ (1). \tag{4}$$

Assume that the input data is perturbed by a bounded and small amount, i.e.,

$$\mathbf{x}_{\epsilon,0} = \mathbf{x}_0 + \epsilon \cdot \mathbf{z},$$

where $\mathbf{z}$ can be either random or adversarial. The proposed CLC-NN adjusts the perturbed state trajectory $\mathbf{x}_{\epsilon,t}$ such that it stays at a minimum distance from the desired manifold $\mathcal{M}_t$ while promoting small magnitudes of controls.

**Intuition.** We use an intuitive example to show how CLC-NN controls the state trajectory of unseen data samples. We create a synthetic binary classification data set with 1500 samples. We train a residual neural network with one hidden layer of dimension 2, and adopt the fast gradient sign method (Goodfellow et al., 2014) to generate adversarial data. Fig. 2 (a) and (b) show the states of clean data (red and blue) and of perturbed data (black and gray) at $t = 0$ and $t = 1$, respectively. The CLC-NN adjusts the state trajectory to reduce the reconstruction loss as shown in Fig. 2 (c) and (d), where lighter background color represents lower reconstruction loss. Comparing Fig. 2 (a) with (c), and Fig. 2 (b) with (d), we see that the perturbed states in Fig. 2 (a) and (b) deviate from the desired state manifold (light green region) and has a high reconstruction loss. Running 1000 iterations of Alg. 1 adjusts the perturbed states and improves the classification accuracy from $86\%$ to $100\%$.

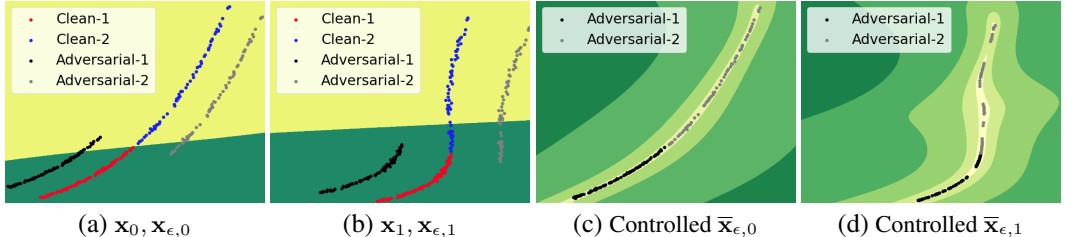

(a) $\mathbf{x}_0, \mathbf{x}_{\epsilon,0}$      (b) $\mathbf{x}_1, \mathbf{x}_{\epsilon,1}$      (c) Controlled $\overline{\mathbf{x}}_{\epsilon,0}$      (d) Controlled $\overline{\mathbf{x}}_{\epsilon,1}$

Figure 2: (a) and (b) show the states of clean data (red and blue) and of perturbed data (black and gray) at the initial and hidden layers respectively. The yellow and green backgrounds represent the two classes predicted by the network. Some of the perturbed data are mis-classified. (c) and (d) show the adjusted states after the proposed close-loop control. The background lightness from light to dark represent the increasing reconstruction loss $\|\mathcal{E}_t(\mathbf{x}_t) - \mathbf{x}_t\|_2^2$.

## 4 IMPLEMENTATION VIA THE PONTRYAGIN'S MAXIMUM PRINCIPLE

**Dynamic Programming for Close-Loop Control (4).** The control problem in Eq. (4) can be solved by the dynamical programming principle (Bellman, 1952). For simplicity we consider one input data sample, and define a value function $V : \mathcal{T} \times \mathbb{R}^d \to \mathbb{R}$ (where $\mathcal{T} := \{0, 1, \ldots, T - 1\}$). Here $V(t, \mathbf{x})$ represents the optimal cost-to-go function of Eq. (4) incurred from time $t$ at state $\mathbf{x}$. One can show that $V(t, \mathbf{x})$ satisfies the dynamic programming principle

$$V(t, \mathbf{x}) = \inf_{\boldsymbol{\pi} \in \boldsymbol{\Pi}} \left[ V(t + 1, \mathbf{x} + f(\mathbf{x}, \boldsymbol{\theta}_t, \boldsymbol{\pi}(\mathbf{x}))) + \mathcal{L}(\mathbf{x}, \boldsymbol{\pi}(\mathbf{x}), \mathcal{E}_t(\cdot)) \right]. \tag{5}$$

Eq. (5) gives a necessary and sufficient condition for optimality of Eq. (4), and it is often solved backward in time by discretizing the entire state space. The state dimension of a modern neural network is at the order of thousands or even higher, therefore, discretizing the state space and directly solving Eq. (5) is intractable for real-world applications due to the curse of dimensionality.

**Solving (5) via the Pontryagin's Maximum Principle.** To overcome the computational challenge, the Pontryagin's Maximum Principle (Kirk, 1970) converts the intractable dynamical programming into two ordinary differential equations and a maximization condition. Instead of computing the control policy $\boldsymbol{\pi}$ of Eq. (5), the Pontryagin's Maximum Principle provides a necessary condition for the optimality with a set of control parameters $[\mathbf{u}_0^*, \cdots, \mathbf{u}_T^*]$. The mean-field Pontryagin's Maximum Principle can be considered when the initial condition is a batch of i.i.d. samples drawn from $\mathcal{D}$. Specifically, we trade the intractable computational complexity with processing time for solving the Hamilton equations and its maximization condition for every newly observed data. To begin with, we define the Hamiltonian $H : \mathcal{T} \times \mathbb{R}^d \times \mathbb{R}^d \times \mathbb{R}^l \times \mathbb{R}^m \to \mathbb{R}$ as

$$H(t, \mathbf{x}_t, \mathbf{p}_{t+1}, \boldsymbol{\theta}_t, \mathbf{u}_t) := \mathbf{p}_{t+1}^T \cdot f(\mathbf{x}_t, \boldsymbol{\theta}_t, \mathbf{u}_t) - \mathcal{L}(\mathbf{x}_t, \mathbf{u}_t, \mathcal{E}_t(\cdot)). \tag{6}$$

Let $\mathbf{x}^*$ denote the corresponding optimally controlled state trajectory. There exists a co-state process $\mathbf{p}^* : [0, T] \to \mathbb{R}^d$ such that the Hamilton's equations

$$\mathbf{x}_{t+1}^* = \nabla_p H(t, \mathbf{x}_t^*, \mathbf{p}_t^*, \boldsymbol{\theta}_t, \mathbf{u}_t^*), \qquad\qquad (\mathbf{x}_0^*, \mathbf{y}) \sim \mathcal{D}, \tag{7}$$

$$\mathbf{p}_t^* = \nabla_x H(t, \mathbf{x}_t^*, \mathbf{p}_{t+1}^*, \boldsymbol{\theta}_t, \mathbf{u}_t^*), \qquad\qquad \mathbf{p}_T^* = \mathbf{0}, \tag{8}$$

are satisfied. The terminal co-state $\mathbf{p}_T = \mathbf{0}$, since we do not consider the terminal loss $\Phi(\mathbf{x}_T, \mathbf{y})$. Moreover, we have the Hamiltonian maximization condition

$$H(t, \mathbf{x}_t^*, \mathbf{p}_t^*, \boldsymbol{\theta}_t, \mathbf{u}_t^*) \geq H(t, \mathbf{x}_t^*, \mathbf{p}_t^*, \boldsymbol{\theta}_t, \mathbf{u}_t), \forall \mathbf{u} \in \mathbb{R}^{d'} \text{ and } \forall t \in \mathcal{T}. \tag{9}$$

Instead of solving Eq. (5) for the optimal control policy $\boldsymbol{\pi}^*(\mathbf{x}_t)$, for a given initial condition, the Pontryagin's Maximum Principle seeks for a open-loop optimal solution such that the global optimum of Eq. (5) is satisfied. The limitation of using the maximum principle is that the control parameters $\mathbf{u}_t^*$ need to be solved for every unseen data to achieve the optimal solution.

**Algorithm Flow.** The numerical implementation of CLC-NN is summarized in Alg. 1. Given a trained network (either from standard or adversarial training) and a set of embedding functions, the controls are initialized as $\mathbf{u}_t = \mathbf{0}, \forall t \in \mathcal{T}$, because adding random initialization weakens the

---

**Algorithm 1:** CLC-NN with the Pontryagin's Maximum Principle.

---

**Input** : Possibly perturbed data $\mathbf{x}_\epsilon$, a trained neural network, embedding functions
$[\mathcal{E}_1, \cdots, \mathcal{E}_{T-1}]$, maxItr (maximum number of iterations).
**Output:** A set of optimal control parameters $\mathbf{u}_0^*, \cdots, \mathbf{u}_{T-1}^*$.

1 **for** $k = 0$ *to* maxItr **do**
2     $J_k = 0$,
3     **for** $t = 0$ *to* $T - 1$ **do**
4        $\mathbf{x}_{t+1,k} = f(\mathbf{x}_{t,k}, \boldsymbol{\theta}_t, \mathbf{u}_{t,k})$,   where $\mathbf{x}_{0,k} = \mathbf{x}_\epsilon$,       ▷ Forward propagation Eq. (7),
5        $J_k = J_k + \mathcal{L}(\mathbf{x}_{t,k}, \mathbf{u}_{t,k}, \mathcal{E}_t(\mathbf{x}_{t,k}))$,          ▷ Objective function Eq. (4),
6     **end for**
7     **for** $t = T$ *to* 1 **do**
8        $\mathbf{p}_{t,k} = \mathbf{p}_{t+1}^T \cdot \nabla_{\mathbf{x}_t} f(\mathbf{x}_{t,k}, \boldsymbol{\theta}_t, \mathbf{u}_{t,k}) - \nabla_{\mathbf{x}_t} \mathcal{L}(\mathbf{x}_{t,k}, \mathbf{u}_{t,k}, \mathcal{E}_t(\mathbf{x}_{t,k}))$,
9        where $\mathbf{p}_{T,k} = \mathbf{0}$,                 ▷ Backward propagation Eq. (8)
10     **end for**
11     **for** $t = 0$ *to* $T - 1$ **do**
12        $\mathbf{u}_{t,k+1} = \mathbf{u}_{t,k} + \left( \mathbf{p}_{t+1,k}^T \cdot \nabla_{\mathbf{u}_t} f(\mathbf{x}_{t,k}, \boldsymbol{\theta}_t, \mathbf{u}_{t,k}) - \nabla_{\mathbf{u}_t} \mathcal{L}(\mathbf{x}_{t,k}, \mathbf{u}_{t,k}, \mathcal{E}_t(\mathbf{x}_{t,k})) \right)$,
13            ▷ Maximization of Hamiltonian Eq. (9) based on Eq. (6) and gradient ascent.
14     **end for**
15 **end for**

---

robustness performance in general, and clean trajectory often does not result in any running loss for the gradient update on the control parameters. In every iteration, a given input $\mathbf{x}_0$ is propagated forwardly with Eq. (7) to obtain all the intermediate hidden states $\mathbf{x}_t$ for all $t$ and to accumulate cost $J$. Eq. (8) backward propagates the co-state $\mathbf{p}_t$ and Eq. (9) maximizes the $t^{th}$ Hamiltonian with current $\mathbf{x}_t$ and $\mathbf{p}_t$ to compute the optimal control parameters $\mathbf{u}_t^*$.

## 5 ERROR ANALYSIS FOR SIMPLIFIED LINEAR CASES

For the ease of analysis, we consider a simplified neural network with linear activation functions:

$$\mathbf{x}_{t+1} = \boldsymbol{\theta}_t(\mathbf{x}_t + \mathbf{u}_t),$$

and reveal why our proposed method can improve robustness in the simplest setting. Given a perturbed data sample $\overline{\mathbf{x}}_{\epsilon,0}$, we denote its perturbation-free counterpart as $\mathbf{x}_0$ so that $\mathbf{z} = \overline{\mathbf{x}}_{\epsilon,0} - \mathbf{x}_0$. We consider a general perturbation where $\mathbf{z}$ is the direct sum of two orthogonal contributions: $\mathbf{z}^\parallel$, which is a perturbation within the data manifold (subspace), and $\mathbf{z}^\perp$, which is a perturbation in the orthogonal complement of the data manifold. This case is general: if we consider adversarial attacks, then the perturbation along the orthogonal complement dominates. In contrast, if we consider random perturbations, then the two perturbations are on the same scale. Our formulation covers both such extreme scenarios, together with intermediate cases.

We use an orthogonal projection as the embedding function such that $\mathcal{E}_t = \mathbf{V}_t^r (\mathbf{V}_t^r)^T$, where $\mathbf{V}_t^r$ is the first $r$ columns of the eigenvectors computed by the Principle Component Analysis on a collection of states $\mathbf{x}_t$. The proposed CLC-NN minimizes $\|\overline{\mathbf{x}}_{\epsilon,t} - \mathbf{x}_t\|_2^2$ by reducing the components of $\mathbf{x}_{\epsilon,t}$ that lie in the the orthogonal complement of $Z_\parallel^t$. The following theorem provides an error estimation between $\overline{\mathbf{x}}_{\epsilon,t}$ and $\mathbf{x}_t$.

**Theorem 1.** *For $t \geq 1$, we have the error estimation*

$$\|\overline{\mathbf{x}}_{\epsilon,t} - \mathbf{x}_t\|_2^2 \leq \|\boldsymbol{\theta}_{t-1} \cdots \boldsymbol{\theta}_0\|_2^2 \cdot \left( \alpha^{2t} \|\mathbf{z}^\perp\|_2^2 + \|\mathbf{z}^\parallel\|_2^2 + \gamma_t \|\mathbf{z}\|_2^2 \left( \gamma_t \alpha^2 (1-\alpha^{t-1})^2 + 2(\alpha - \alpha^t) \right) \right), \quad (10)$$

*where* $\gamma_t := \max_{s \leq t} \left( 1 + \kappa(\overline{\boldsymbol{\theta}}_s)^2 \right) \|\mathbf{I} - \overline{\boldsymbol{\theta}}_s^T \overline{\boldsymbol{\theta}}_s\|_2$, *and* $\alpha = \frac{c}{1+c}$, *$c$ represents the control regularization. In particular, the equality*

$$\|\overline{\mathbf{x}}_{\epsilon,t} - \mathbf{x}_t\|_2^2 = \alpha^{2t} \|\mathbf{z}^\perp\|_2^2 + \|\mathbf{z}^\parallel\|_2^2, \quad (11)$$

*holds when all $\boldsymbol{\theta}_t$ are orthogonal.*

The detailed derivation is presented in Appendix A. Let us summarize the insights from Theorem 1.

Table 1: Experimental results on ResNet-20 from standard training.

| Dataset | $\epsilon$ | Accuracy: original model without CLC / CLC-NN + Linear / CLC-NN + Nonlinear | | | | |
| --- | --- | --- | --- | --- | --- | --- |
| | | Type of input perturbations | | | | |
| | | None | Manifold | FGSM | PGD | CW |
| CIFAR-10 | 2 | 92 / 88 / 89 | 24 / 79 / 82 | 21 / 56 / 56 | 0 / 50 / 50 | 8 / 75 / 79 |
| | 4 | | 5 / 78 / 81 | 11 / 40 / 30 | 0 / 31 / 19 | 0 / 75 / 79 |
| | 8 | | 1 / 78 / 81 | 8 / 20 / 12 | 0 / 11 / 2 | 0 / 76 / 79 |
| CIFAR-100 | 2 | 69 / 60 / 58 | 9 / 51 / 52 | 9 / 25 / 23 | 0 / 17 / 22 | 4 / 47 / 49 |
| | 4 | | 3 / 50 / 52 | 5 / 15 / 9 | 0 / 6 / 4 | 1 / 47 / 49 |
| | 8 | | 2 / 50 / 52 | 4 / 9 / 5 | 0 / 1 / 0 | 0 / 47 / 49 |

- The above error estimation is general for any input perturbation. It shows the working principle behind the proposed CLC-NN on controlling the perturbation that lies in the orthogonal complement of input subspace ($\mathbf{z}^{\perp}$).

- The above error estimation improves as the control regularization $c$ goes to $0$ (so $\alpha \to 0$). It is not the sharpest possible as it relies on a greedily optimal control at each layer. The globally optimal control defined by the Ricatti equation may achieve a lower loss when $c \neq 0$.

- When the dimension of embedding subspace $r$ decreases, our control becomes more effective in reducing $\|\overline{\mathbf{x}}_{\epsilon,t} - \mathbf{x}_t\|_2^2$. This means that the control approach works the best when the data is constrained on a low dimensional manifold, which is consistent with the manifold hypothesis. In particular, observe that as $r \to 0$, $\|\mathbf{z}^{\|}\|_2^2 \to 0$

- The obtained upper bound is tight: the estimated upper bound becomes the actual error if all the forward propagation layers are orthogonal matrices.

## 6 NUMERICAL EXPERIMENTS

We test our proposed CLC-NN framework under various input data perturbations. Here we briefly summarize our experimental settings, and we refer readers to Appendix B for the details.

- **Original Networks without Close-Loop Control.** We choose residual neural networks (He et al., 2016) with ReLU activation functions as our target for close-loop control. In order to show that CLC-NN can improve the robustness in various settings, we consider networks from both standard and adversarial trainings. We consider multiple adversarial training methods: fast gradient sign method (**FGSM**) (Goodfellow et al., 2014), project gradient descent (**PGD**) (Madry et al., 2017), and the Label smoothing training (**Label Smooth**) (Hazan et al., 2017).

- **Input Perturbations.** In order to test our CLC-NN framework, we perturb the input data within a radius of $\epsilon$ with $\epsilon = 2, 4$ and $8$ respectively. We consider various perturbations, including non-adversarial perturbations with the manifold-based attack (Jalal et al., 2017) (**Manifold**), as well as some adversarial attacks such as **FGSM**, **PGD** and the **CW** methods (Carlini & Wagner, 2017).

- **CLC-NN Implementations.** We consider both linear and nonlinear embedding in our close-loop control. Specifically, we employ a principal component analysis with a $1\%$ truncation error for linear embedding, and convolutional auto-encoders for nonlinear embedding. We use Adam (Kingma & Ba, 2014) to maximize the Hamiltonian function (9) and *keep the same hyperparameters (learning rate, maximum iterations) for each model against all perturbations.*

**Result Summary:** Table 1 and Table 2 show the results for both CIFAR-10 and CIFAR-100 datasets on some neural networks from both standard training and adversarial training respectively.

- **CLC-NN significantly improves the robustness of neural networks from standard training.** Table 1 shows that the baseline network trained on a clean data set becomes completely vulnerable (with almost $0\%$ accuracy) under PGD and CW attacks. Our CLC-NN improves its accuracy to nearly $40\%$ and $80\%$ under PGD and CW attacks respectively. The accuracy under FGSM attacks has almost been doubled by our CLC-NN method. The accuracy on clean data is slightly decreased because the lower-dimensional embedding functions cannot exactly capture $Z_{\|}$ or $\mathcal{M}$.

- **CLC-NN further improves the robustness of adversarially trained networks.** Table 2 shows that while an adversarially trained network is inherently robust against certain types of perturbations, CLC-NN strengthens its robustness significantly against various perturbations. For in-

Table 2: Experimental results on robustly trained networks.

| Dataset | Training methods | $\epsilon$ | Accuracy: Robustly trained models / CLC-NN + Linear Embedding | | | | |
|---|---|---|---|---|---|---|---|
| | | | | Type of input perturbations | | | |
| | | | None | Manifold | FGSM | PGD | CW |
| CIFAR-10 | FGSM | 2 | 92 / 90 | 43 / 82 | 90 / 87 | 1 / 60 | 14 / 81 |
| | | 4 | | 17 / 81 | 95 / 90 | 0 / 36 | 2 / 81 |
| | | 8 | | 3 / 80 | 96 / 93 | 0 / 10 | 0 / 81 |
| | Label Smooth | 2 | 93 / 89 | 42 / 78 | 57 / 69 | 10 / 60 | 29 / 76 |
| | | 4 | | 20 / 77 | 51 / 61 | 1 / 49 | 11 / 76 |
| | | 8 | | 6 / 77 | 40 / 47 | 0 / 26 | 2 / 76 |
| | PGD | 2 | 83 / 80 | 80 / 78 | 75 / 73 | 74 / 73 | 75 / 75 |
| | | 4 | | 78 / 76 | 66 / 66 | 63 / 65 | 66 / 71 |
| | | 8 | | 74 / 73 | 49 / 51 | 40 / 46 | 48 / 64 |
| CIFAR-100 | FGSM | 2 | 69 / 66 | 21 / 55 | 59 / 54 | 1 / 27 | 9 / 57 |
| | | 4 | | 8 / 54 | 66 / 55 | 0 / 9 | 2 / 57 |
| | | 8 | | 3 / 53 | 67 / 56 | 0 / 1 | 0 / 57 |
| | Label Smooth | 2 | 69 / 62 | 10 / 43 | 16 / 28 | 1 / 19 | 5 / 46 |
| | | 4 | | 3 / 42 | 12 / 19 | 0 / 9 | 1 / 46 |
| | | 8 | | 2 / 42 | 8 / 11 | 0 / 2 | 0 / 46 |
| | PGD | 2 | 58 / 52 | 52 / 50 | 46 / 43 | 45 / 43 | 45 / 47 |
| | | 4 | | 49 / 48 | 36 / 36 | 34 / 35 | 35 / 43 |
| | | 8 | | 43 / 45 | 22 / 24 | 16 / 21 | 21 / 40 |

Table 3: Accuracy comparision of CLC-NN and reactive defense in Eq. (12), with $\epsilon_{attack} = 2 \,/\, 4 \,/\, 8$. Here "+" (or "-") indicates how much CLC-NN outperforms (or underperforms) reactive defense.

| Method | Type of input perturbations | | | | |
|---|---|---|---|---|---|
| | None | Manifold | FGSM | PGD | CW |
| CIFAR-10 | -3 | +47 / +63 / +66 | +27 / +20 / +13 | +43 / +35 / +25 | +66 / +76 / +77 |
| CIFAR-100 | +1 | +34 / +37 / +38 | +22 / 0 / +9 | +44 / +30 / +11 | +37 / +30 / +16 |

stance, CLC-NN improves the accuracy of an FGSM trained network under PGD and CW attacks by a maximum of $59\%$ and $81\%$, respectively.

- **The robustness improvement of adversarially trained networks is less significant.** This is expected because the trajectory of perturbed data lies on the embedding subspace $Z_\parallel$ if that data sample has been used in adversarial training. However, our experiments show that applying CLC-NN to adversarially trained networks can achieve the best performance under most attacks.

**Comparison with PixelDefend (Song et al., 2017).** Our method achieves similar performance on CIFAR-10 with slightly different experimental setting. Specifically, PixelDefend improved the robustness of a normally trained 62-layer ResNet from $0\%$ to $78\%$ against CW attack. Our proposed CLC-NN improves the robustness of a 20-layer ResNet from $0\%$ to $81\%$ against CW attacks. Furthermore, we show that CLC-NN is robust against the manifold-based attack. No result was reported for CIFAR-100 in Song et al. (2017).

**Comparison with Reactive Defense** Reactive defenses can be understood as only applying a control at the initial condition of a dynamical system. Specifically, reactive defense equipped with linear embedding admits the following dynamics:

$$\bar{\mathbf{x}}_{t+1} = f(\bar{\mathbf{x}}_t, \boldsymbol{\theta}_t), \quad s.t. \ \bar{\mathbf{x}}_0 = \mathbf{V}_0^r (\mathbf{V}_0^r)^T \mathbf{x}_{\epsilon,0}. \tag{12}$$

By contrast, CLC-NN controls all hidden states and results in a decreasing error as the number of layers $T$ increases (c.f. Theorem 1). To quantitatively compare CLC-NN with reactive defense, we implement them with the same linear embedding functions and against all perturbations. In Table 3, CLC-NN outperforms reactive defense in almost all cases except that their performances are case-dependent on clean data.

## 7 CONCLUSION

We have proposed a close-loop control formulation to improve the robustness of neural networks. We have studied the embedding of state trajectory during forward propagation to define the optimal control objective function. The numerical experiments have shown that our method can improve

the robustness of a trained neural network against various perturbations. We have provided an error estimation for the proposed method in the linear case. Our current implementation uses the Pontryagin's Maximum Principle and an online iterative algorithm to overcome the intractability of solving a dynamical programming. This online process adds extra inference time. In the future, we plan to show the theoretical analysis for the nonlinear embedding case.

**Acknowledgement**    Zhuotong Chen and Zheng Zhang are supported by NSF CAREER Award No. 1846476 and NSF CCF No. 1817037. Qianxiao Li is supported by the start-up grant under the NUS PYP programme.

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

## A  APPENDIX A  ERROR ESTIMATION FOR THE PROPOSED CLC-NN

**Preliminaries**  We define the performance index at time $t$ as

$$J(\mathbf{x}_t, \mathbf{u}_t) = \frac{1}{2}\|\mathbf{Q}_t(\mathbf{x}_t + \mathbf{u}_t)\|_2^2 + \frac{c}{2}\|\mathbf{u}_t\|_2^2, \tag{13}$$

where $\mathbf{Q}_t = \mathbf{I} - \mathbf{V}_t^r(\mathbf{V}_t^r)^T$, $\mathbf{V}_t^r$ is the linear projection matrix at time $t$ with only its first $r$ principle components corresponding to the largest $r$ eigenvalues. The optimal feedback control is defined as

$$\mathbf{u}_t^*(\mathbf{x}_t) = \arg\min_{\mathbf{u}_t} J(\mathbf{x}_t, \mathbf{u}_t),$$

due to the linear system and quadratic performance index, the optimal feedback control admits an analytic solution by taking the gradient of performance index (Eq. (13)) and setting it to $\mathbf{0}$.

$$\nabla_{\mathbf{u}} J(\mathbf{x}_t, \mathbf{u}_t) = \nabla_{\mathbf{u}}\left(\frac{1}{2}\|\mathbf{Q}_t(\mathbf{x}_t + \mathbf{u}_t)\|_2^2 + \frac{c}{2}\|\mathbf{u}_t\|_2^2\right),$$
$$= \mathbf{Q}_t^T\mathbf{Q}_t\mathbf{x}_t + \mathbf{Q}_t^T\mathbf{Q}_t\mathbf{u}_t + c \cdot \mathbf{u}_t,$$

which leads to the analytic solution of $\mathbf{u}_t^*(\mathbf{x}_t)$ as

$$\mathbf{u}_t^*(\mathbf{x}_t) = -(c \cdot \mathbf{I} + \mathbf{Q}_t^T\mathbf{Q}_t)^{-1}\mathbf{Q}_t^T\mathbf{Q}_t\mathbf{x}_t. \tag{14}$$

The above analytic control solution $\mathbf{u}_t^*$ optimizes the performance index instantly at time step $t$, the error measured by Eq. (13) for the dynamical programming solution $\bar{\mathbf{x}}_{\epsilon,t}$ must be smaller or equal

to the state trajectory equipped with $\mathbf{u}_t^*$ define by Eq. (14), which gives a guaranteed upper bound for the error estimation of the dynamic programming solution.

We define the feedback gain matrix $\mathbf{K}_t = (c \cdot \mathbf{I} + \mathbf{Q}_t^T \mathbf{Q}_t)^{-1} \mathbf{Q}_t^T \mathbf{Q}_t$. Thus, the one-step optimal feedback control can be represented as $\mathbf{u}_t^* = -\mathbf{K}_t \mathbf{x}_t$.

The difference between the controlled system applied with perturbation at initial condition and the uncontrolled system without perturbation is shown

$$\begin{aligned}
\overline{\mathbf{x}}_{\epsilon,t+1} - \mathbf{x}_{t+1} &= \boldsymbol{\theta}_t(\overline{\mathbf{x}}_{\epsilon,t} + \mathbf{u}_t - \mathbf{x}_t), \\
&= \boldsymbol{\theta}_t(\overline{\mathbf{x}}_{\epsilon,t} - \mathbf{K}_t \overline{\mathbf{x}}_{\epsilon,t} - \mathbf{x}_t).
\end{aligned} \tag{15}$$

The control objective is to minimize the state components that span the orthogonal complement of the data manifold $(\mathbf{I} - \mathbf{V}_t^r(\mathbf{V}_t^r)^T)$, when the input data to feedback control only stays in the state manifold, such that $\|(\mathbf{I} - \mathbf{V}_t^r(\mathbf{V}_t^r)^T)\mathbf{x}_t\|_2^2 = 0$, the feedback control $\mathbf{K}_t \mathbf{x}_t = \mathbf{0}$. The state difference of Eq. (15) can be further shown by adding a $\mathbf{0}$ term of $(\boldsymbol{\theta}_t \mathbf{K}_t \mathbf{x}_t)$

$$\begin{aligned}
\overline{\mathbf{x}}_{\epsilon,t+1} - \mathbf{x}_{t+1} &= \boldsymbol{\theta}_t(\mathbf{I} - \mathbf{K}_t)\overline{\mathbf{x}}_{\epsilon,t} - \boldsymbol{\theta}_t \mathbf{x}_t + \boldsymbol{\theta}_t \mathbf{K}_t \mathbf{x}_t, \\
&= \boldsymbol{\theta}_t(\mathbf{I} - \mathbf{K}_t)(\overline{\mathbf{x}}_{\epsilon,t} - \mathbf{x}_t).
\end{aligned} \tag{16}$$

In the following, we show a transformation on the control dynamic term $(\mathbf{I} - \mathbf{K}_t)$ based on its definition.

**Lemma 1.** *For $t \geq 0$, we have*

$$\mathbf{I} - \mathbf{K}_t = \alpha \cdot \mathbf{I} + (1 - \alpha) \cdot \mathbf{P}_t,$$

*where $\mathbf{P}_t := \mathbf{V}_t^r(\mathbf{V}_t^r)^T$, which is the orthogonal projection onto $Z_\|^t$, and $\alpha := \frac{c}{1+c}$ such that $\alpha \in [0, 1]$.*

*Proof.* Recall that $\mathbf{K}_t = (c \cdot \mathbf{I} + \mathbf{Q}_t^T \mathbf{Q}_t)^{-1} \mathbf{Q}_t^T \mathbf{Q}_t$, and $\mathbf{Q}_t = \mathbf{I} - \mathbf{V}_t^r(\mathbf{V}_t^r)^T$, $\mathbf{Q}_t$ can be diagonalized as following

$$\mathbf{Q}_t = \mathbf{V}_t \begin{bmatrix} 0 & 0 & \cdots & 0 & 0 \\ 0 & 0 & \cdots & 0 & 0 \\ \vdots & \vdots & \ddots & 0 & 0 \\ 0 & 0 & \cdots & 1 & 0 \\ 0 & 0 & \cdots & 0 & 1 \end{bmatrix} \mathbf{V}_t^T,$$

where the first $r$ diagonal elements have common value of $0$ and the last $(d - r)$ diagonal elements have common value of $1$. Furthermore, the feedback gain matrix $\mathbf{K}_t$ can be diagonalized as

$$\mathbf{K}_t = \mathbf{V}_t \begin{bmatrix} 0 & 0 & \cdots & 0 & 0 \\ 0 & 0 & \cdots & 0 & 0 \\ \vdots & \vdots & \ddots & 0 & 0 \\ 0 & 0 & \cdots & \frac{1}{1+c} & 0 \\ 0 & 0 & \cdots & 0 & \frac{1}{1+c} \end{bmatrix} \mathbf{V}_t^T,$$

where the last $(d - r)$ diagonal elements have common value of $\frac{1}{1+c}$. The control term $(\mathbf{I} - \mathbf{K}_t)$ thus can be represented as

$$\mathbf{I} - \mathbf{K}_t = \mathbf{V}_t \begin{bmatrix} 1 & 0 & \cdots & 0 & 0 \\ 0 & 1 & \cdots & 0 & 0 \\ \vdots & \vdots & \ddots & 0 & 0 \\ 0 & 0 & \cdots & \frac{c}{1+c} & 0 \\ 0 & 0 & \cdots & 0 & \frac{1}{1+c} \end{bmatrix} \mathbf{V}_t^T,$$

where the first $r$ diagonal elements have common value of $1$ and the last $(d - r)$ diagonal elements have common value of $\frac{c}{1+c}$. By denoting the projection of first $r$ columns as $\mathbf{V}_t^r$ and last $(d - r)$ columns as $\hat{\mathbf{V}}_t^r$, it can be further shown as

$$\begin{aligned}
\mathbf{I} - \mathbf{K}_t &= \mathbf{V}_t^r(\mathbf{V}_t^r)^T + \frac{c}{1+c}(\hat{\mathbf{V}}_t^r(\hat{\mathbf{V}}_t^r)^T), \\
&= \mathbf{P}_t + \alpha(\mathbf{I} - \mathbf{P}_t), \\
&= \alpha \cdot \mathbf{I} + (1 - \alpha) \cdot \mathbf{P}_t.
\end{aligned}$$

$\square$

**Oblique Projections.** Let $\mathbf{P}$ be a linear operator on $\mathbb{R}^d$,

- We say that $\mathbf{P}$ is an projection if $\mathbf{P}^2 = \mathbf{P}$.
- $\mathbf{P}$ is an orthogonal projection if $\mathbf{P} = \mathbf{P}^T = \mathbf{P}^2$.
- If $\mathbf{P}^2 = \mathbf{P}$ but $\mathbf{P} \neq \mathbf{P}^T$, it is called an oblique projection.

**Proposition 2.** *For a projection* $\mathbf{P}$,

1. *If* $\mathbf{P}$ *is an orthogonal projection, then* $\|\mathbf{P}\|_2 = 1$.

2. *If* $\mathbf{P}$ *is an oblique projection, then* $\|\mathbf{P}\|_2 > 1$.

3. *If* $\mathbf{P}$, $\mathbf{Q}$ *are two projections such that* $range(\mathbf{P}) = range(\mathbf{Q})$, *then* $\mathbf{PQ} = \mathbf{Q}$ *and* $\mathbf{QP} = \mathbf{P}$.

4. *If* $\mathbf{P}$ *is a projection, then* $rank(\mathbf{P}) = Tr(\mathbf{P})$. *Furthermore, if* $\mathbf{P}$ *is an orthogonal projection, then* $rank(\mathbf{P}) = \|\mathbf{P}\|_F^2 = Tr(\mathbf{P}\mathbf{P}^T)$.

Define for $t \geq 0$

$$\begin{cases} \mathbf{P}_t^0 := \mathbf{P}_t, \\ \mathbf{P}_t^{(s+1)} := \boldsymbol{\theta}_{t-s-1}^{-1} \mathbf{P}_t^s \boldsymbol{\theta}_{t-s-1}, \quad s = 0, 1, \ldots, t-1, \end{cases}$$

**Lemma 3.** *Let* $\mathbf{P}_t^s$ *be defined as above for* $0 \leq s \leq t$. *Then*

1. $\mathbf{P}_t^s$ *is a projection.*

2. $\mathbf{P}_t^s$ *is a projection onto* $Z_\|^{t-s}$, *i.e.* $range(\mathbf{P}_t^s) = Z_\|^{t-s}$.

3.
$$\|\mathbf{P}_t^s\|_F^2 \leq \kappa(\boldsymbol{\theta}_{t-1}\boldsymbol{\theta}_{t-2}\ldots\boldsymbol{\theta}_{t-s})^2 \cdot r,$$
*where* $\kappa(\mathbf{A})$ *is the condition number of* $\mathbf{A}$, *i.e.* $\kappa(\mathbf{A}) = \|\mathbf{A}\|_2 \cdot \|\mathbf{A}^{-1}\|_2$, *and* $r = rank(Z_\|^0) = rank(Z_\|^1) = \ldots = rank(Z_\|^t)$.

*Proof.*

1. We prove it by induction on $s$ for each $t$. For $s = 0$, $\mathbf{P}_t^0 = \mathbf{P}_t$, which is a projection by its definition. Suppose it is true for $s$ such that $\mathbf{P}_t^s = \mathbf{P}_t^s \mathbf{P}_t^s$, then for $(s+1)$,

$$\begin{aligned} (\mathbf{P}_t^{s+1})^2 &= \left(\boldsymbol{\theta}_{t-s-1}^{-1} \mathbf{P}_t^s \boldsymbol{\theta}_{t-s-1}\right)^2, \\ &= \boldsymbol{\theta}_{t-s-1}^{-1} \left(\mathbf{P}_t^s\right)^2 \boldsymbol{\theta}_{t-s-1}, \\ &= \boldsymbol{\theta}_{t-s-1}^{-1} \mathbf{P}_t^s \boldsymbol{\theta}_{t-s-1}, \\ &= \mathbf{P}_t^{s+1}. \end{aligned}$$

2. We prove it by induction on $s$ for each $t$. For $s = 0$, $\mathbf{P}_t^0 = \mathbf{P}_t$, which is the orthogonal projection onto $Z_\|^t$. Suppose that it is true for $s$ such that $\mathbf{P}_t^s$ is a projection onto $Z_\|^{t-s}$, then for $(s+1)$, $\mathbf{P}_t^{s+1} = \boldsymbol{\theta}_{t-s-1}^{-1} \mathbf{P}_t^s \boldsymbol{\theta}_{t-s-1}$, which implies

$$\begin{aligned} range(\mathbf{P}_t^{s+1}) &= range(\boldsymbol{\theta}_{t-s-1}^{-1} \mathbf{P}_t^s), \\ &= \{\boldsymbol{\theta}_{t-s-1}^{-1}\mathbf{x} : \mathbf{x} \in Z_\|^{t-s}\}, \\ &= Z_\|^{t-s-1}. \end{aligned}$$

3. We use the inequalities $\|\mathbf{AB}\|_F \leq \|\mathbf{A}\|_2\|\mathbf{B}\|_F$, and $\|\mathbf{AB}\|_F \leq \|\mathbf{A}\|_F\|\mathbf{B}\|_2$. By the definition of $\mathbf{P}_t^s$,
$$\mathbf{P}_t^s = \left(\boldsymbol{\theta}_{t-1}\boldsymbol{\theta}_{t-2}\cdots\boldsymbol{\theta}_{t-s}\right)^{-1} \mathbf{P}_t^0 \left(\boldsymbol{\theta}_{t-1}\boldsymbol{\theta}_{t-2}\cdots\boldsymbol{\theta}_{t-s}\right),$$
we have the following

$$\begin{aligned} \|\mathbf{P}_t^s\|_F^2 &\leq \|\left(\boldsymbol{\theta}_{t-1}\boldsymbol{\theta}_{t-2}\cdots\boldsymbol{\theta}_{t-s}\right)^{-1}\|_2^2 \cdot \|\left(\boldsymbol{\theta}_{t-1}\boldsymbol{\theta}_{t-2}\cdots\boldsymbol{\theta}_{t-s}\right)\|_2^2 \cdot \|\mathbf{P}_t^0\|_F^2, \\ &\leq \kappa(\boldsymbol{\theta}_{t-1}\boldsymbol{\theta}_{t-2}\cdots\boldsymbol{\theta}_{t-s})^2 \cdot r, \hspace{4cm} \text{Lemma 2(4).} \end{aligned}$$

$\square$

The following Lemma uses the concept of oblique projection to show a recursive relationship to project any $t^{th}$ state space of Eq. (16) back to the input data space.

**Lemma 4.** *Define for $0 \le s \le t$,*

$$\mathbf{G}_t^s := \alpha \cdot \mathbf{I} + (1 - \alpha)\mathbf{P}_t^s.$$

*Then, Eq. (16) can be written as*

$$\overline{\mathbf{x}}_{\epsilon,t} - \mathbf{x}_t = (\boldsymbol{\theta}_{t-1}\boldsymbol{\theta}_{t-2}\cdots\boldsymbol{\theta}_0)(\mathbf{G}_{t-1}^{t-1}\mathbf{G}_{t-2}^{t-2}\cdots\mathbf{G}_0^0)(\overline{\mathbf{x}}_{\epsilon,0} - \mathbf{x}_0), \quad t \ge 1.$$

*Proof.* We prove it by induction on $t$. For $t = 1$, by the definition of $\mathbf{G}_t^s$ and transformation from Lemma 1,

$$
\begin{aligned}
\overline{\mathbf{x}}_{\epsilon,1} - \mathbf{x}_1 &= \boldsymbol{\theta}_0(\mathbf{I} - \mathbf{K}_0)(\overline{\mathbf{x}}_{\epsilon,0} - \mathbf{x}_0), & \text{Eq. (16)}, \\
&= \boldsymbol{\theta}_0(\alpha \cdot \mathbf{I} + (1 - \alpha) \cdot \mathbf{P}_0)(\overline{\mathbf{x}}_{\epsilon,0} - \mathbf{x}_0), & \text{Lemma 1}, \\
&= \boldsymbol{\theta}_0\mathbf{G}_0^0(\overline{\mathbf{x}}_{\epsilon,0} - \mathbf{x}_0).
\end{aligned}
$$

Suppose that it is true for $(\overline{\mathbf{x}}_{\epsilon,t} - \mathbf{x}_t)$, by using Eq. (16) and Lemma 1, we have

$$
\begin{aligned}
\overline{\mathbf{x}}_{\epsilon,t+1} - \mathbf{x}_{t+1} &= \boldsymbol{\theta}_t(\mathbf{I} - \mathbf{K}_t)(\overline{\mathbf{x}}_{\epsilon,t} - \mathbf{x}_t), & \text{Eq. (16)}, \\
&= \boldsymbol{\theta}_t(\alpha \cdot \mathbf{I} - (1 - \alpha) \cdot \mathbf{P}_t)(\overline{\mathbf{x}}_{\epsilon,t} - \mathbf{x}_t), & \text{Lemma 1}, \\
&= \boldsymbol{\theta}_t\mathbf{G}_t^0(\boldsymbol{\theta}_{t-1}\boldsymbol{\theta}_{t-2}\cdots\boldsymbol{\theta}_0)(\mathbf{G}_{t-1}^{t-1}\mathbf{G}_{t-2}^{t-2}\cdots\mathbf{G}_0^0)(\overline{\mathbf{x}}_{\epsilon,0} - \mathbf{x}_0). & (17)
\end{aligned}
$$

Recall the definitions of $\mathbf{P}_t^{(s+1)} := \boldsymbol{\theta}_{t-s-1}^{-1}\mathbf{P}_t^s\boldsymbol{\theta}_{t-s-1}$, and $\mathbf{G}_t^s := \alpha \cdot \mathbf{I} + (1 - \alpha)\mathbf{P}_t^s$, we have the following

$$
\begin{aligned}
\mathbf{G}_t^{s+1} &= \alpha \cdot \mathbf{I} + (1 - \alpha) \cdot \mathbf{P}_t^{(s+1)}, \\
&= \alpha \cdot \mathbf{I} + (1 - \alpha) \cdot \boldsymbol{\theta}_{t-s-1}^{-1}\mathbf{P}_t^s\boldsymbol{\theta}_{t-s-1}, \\
&= \boldsymbol{\theta}_{t-s-1}^{-1}\big(\alpha \cdot \mathbf{I} + (1 - \alpha) \cdot \mathbf{P}_t^s\big)\boldsymbol{\theta}_{t-s-1}, \\
&= \boldsymbol{\theta}_{t-s-1}^{-1}\mathbf{G}_t^s\boldsymbol{\theta}_{t-s-1},
\end{aligned}
$$

which results in the equality for the oblique projections. Furthermore,

$$\boldsymbol{\theta}_{t-s-1}\mathbf{G}_t^{(s+1)} = \mathbf{G}_t^s\boldsymbol{\theta}_{t-s-1}.$$

Applying the above to Eq. (17) results in

$$
\begin{aligned}
\overline{\mathbf{x}}_{\epsilon,t+1} - \mathbf{x}_{t+1} &= \boldsymbol{\theta}_t\mathbf{G}_t^0(\boldsymbol{\theta}_{t-1}\boldsymbol{\theta}_{t-2}\cdots\boldsymbol{\theta}_0)(\mathbf{G}_{t-1}^{t-1}\mathbf{G}_{t-2}^{t-2}\cdots\mathbf{G}_0^0)(\overline{\mathbf{x}}_{\epsilon,0} - \mathbf{x}_0), \\
&= (\boldsymbol{\theta}_t\boldsymbol{\theta}_{t-1})\mathbf{G}_t^1(\boldsymbol{\theta}_{t-2}\boldsymbol{\theta}_{t-3}\cdots\boldsymbol{\theta}_0)(\mathbf{G}_{t-1}^{t-1}\mathbf{G}_{t-2}^{t-2}\cdots\mathbf{G}_0^0)(\overline{\mathbf{x}}_{\epsilon,0} - \mathbf{x}_0), \\
&= (\boldsymbol{\theta}_t\boldsymbol{\theta}_{t-1}\boldsymbol{\theta}_{t-2})\mathbf{G}_t^2(\boldsymbol{\theta}_{t-3}\boldsymbol{\theta}_{t-4}\cdots\boldsymbol{\theta}_0)(\mathbf{G}_{t-1}^{t-1}\mathbf{G}_{t-2}^{t-2}\cdots\mathbf{G}_0^0)(\overline{\mathbf{x}}_{\epsilon,0} - \mathbf{x}_0), \\
&= (\boldsymbol{\theta}_t\boldsymbol{\theta}_{t-1}\cdots\boldsymbol{\theta}_0)(\mathbf{G}_t^t\mathbf{G}_{t-1}^{t-1}\cdots\mathbf{G}_0^0)(\overline{\mathbf{x}}_{\epsilon,0} - \mathbf{x}_0).
\end{aligned}
$$

$\square$

**Lemma 5.** *Let*

$$\mathbf{F}_t := \mathbf{G}_{t-1}^{(t-1)}\mathbf{G}_{t-2}^{(t-2)}\cdots\mathbf{G}_0^0, \quad t \ge 1.$$

*Then,*

$$\mathbf{F}_t = \alpha^t \cdot \mathbf{I} + (1 - \alpha)\sum_{s=0}^{t-1}\alpha^s\mathbf{P}_s^s.$$

*Proof.* We prove it by induction on $t$. Recall the definition of $\mathbf{G}_t^s := \alpha \cdot \mathbf{I} + (1 - \alpha) \cdot \mathbf{P}_t^s$. When $t = 1$,

$$\mathbf{F}_1 = \mathbf{G}_0^0 = \alpha \cdot \mathbf{I} + (1 - \alpha) \cdot \mathbf{P}_0^0.$$

Suppose that it is true for $t$ such that

$$\mathbf{F}_t = \mathbf{G}_{t-1}^{(t-1)}\mathbf{G}_{t-2}^{(t-2)}\cdots\mathbf{G}_0^0 = \alpha^t \cdot \mathbf{I} + (1 - \alpha)\sum_{s=0}^{t-1}\alpha^s\mathbf{P}_s^s,$$

for $(t+1)$,

$$
\begin{aligned}
\mathbf{F}_{t+1} &= \mathbf{G}_t^t F(t), \\
&= (\alpha \cdot \mathbf{I} + (1-\alpha) \cdot \mathbf{P}_t^t)\mathbf{F}_t, \\
&= (\alpha \cdot \mathbf{I} + (1-\alpha) \cdot \mathbf{P}_t^t)(\alpha^t \cdot \mathbf{I} + (1-\alpha)\sum_{s=0}^{t-1} \alpha^s \mathbf{P}_s^s), \\
&= \alpha^{t+1} \cdot \mathbf{I} + \alpha^t(1-\alpha)\mathbf{P}_t^t + (1-\alpha)^2\sum_{s=0}^{t-1} \alpha^s \cdot \mathbf{P}_t^t \mathbf{P}_s^s + \alpha(1-\alpha)\sum_{s=0}^{t-1} \alpha^s \cdot \mathbf{P}_s^s.
\end{aligned}
$$

Recall Lemma 3, $range(\mathbf{P}_t^t) = range(\mathbf{P}_s^s) = Z_{\parallel}^0$. According to Proposition 2 (3), $\mathbf{P}_t^t \mathbf{P}_s^s = \mathbf{P}_s^s$. Hence,

$$
\begin{aligned}
\mathbf{F}_{t+1} &= \alpha^{t+1} \cdot \mathbf{I} + \alpha^t(1-\alpha) \cdot \mathbf{P}_t^t + (1-\alpha)\sum_{s=0}^{t-1} \alpha^s \cdot \mathbf{P}_s^s, \\
&= \alpha^{t+1} \cdot \mathbf{I} + (1-\alpha)\sum_{s=0}^{t} \alpha^s \cdot \mathbf{P}_s^s.
\end{aligned}
$$

$\square$

**Lemma 6.** *Let $\mathbf{V} \in \mathbb{R}^{d \times r}$ be a matrix whose columns are an orthogonal basis for a subspace $\mathcal{D}$, and $\boldsymbol{\theta} \in \mathbb{R}^{d \times d}$ be invertible. Let $\mathbf{P} = \mathbf{V}\mathbf{V}^T$ be the orthogonal projection onto $\mathcal{D}$. Denote by $\hat{\mathbf{P}}$ the orthogonal projection onto $\boldsymbol{\theta}\mathcal{D} := \{\boldsymbol{\theta}\mathbf{x} : \mathbf{x} \in \mathcal{D}\}$. Then*

1. *$\boldsymbol{\theta}^{-1}\hat{\mathbf{P}}\boldsymbol{\theta}$ is an oblique projection onto $\mathcal{D}$.*

2. *$\|\boldsymbol{\theta}^{-1}\hat{\mathbf{P}}\boldsymbol{\theta} - \mathbf{P}\|_2 \leq \big(1 + \kappa(\boldsymbol{\theta})^2\big) \cdot \|\mathbf{I} - \boldsymbol{\theta}^T\boldsymbol{\theta}\|_2.$*

*In general, the last inequality shows that $\boldsymbol{\theta}^{-1}\hat{\mathbf{P}}\boldsymbol{\theta} = \mathbf{P}$, if $\boldsymbol{\theta}$ is orthogonal.*

*Proof.*

1. $(\boldsymbol{\theta}^{-1}\hat{\mathbf{P}}\boldsymbol{\theta})^2 = \boldsymbol{\theta}^{-1}\hat{\mathbf{P}}^2\boldsymbol{\theta} = \boldsymbol{\theta}^{-1}\hat{\mathbf{P}}\boldsymbol{\theta}$, therefore, $\boldsymbol{\theta}^{-1}\hat{\mathbf{P}}\boldsymbol{\theta}$ is an projection.

2. Since $\hat{\mathbf{P}}$ is orthogonal projection onto the row space of $\boldsymbol{\theta}\mathbf{V}$, then

$$
\begin{aligned}
\hat{\mathbf{P}} &= \boldsymbol{\theta}\mathbf{V}\big[(\boldsymbol{\theta}\mathbf{V})^T(\boldsymbol{\theta}\mathbf{V})\big]^{-1}(\boldsymbol{\theta}\mathbf{V})^T, \\
&= \boldsymbol{\theta}\mathbf{V}\big[\mathbf{V}^T\boldsymbol{\theta}^T\boldsymbol{\theta}\mathbf{V}\big]^{-1}\mathbf{V}^T\boldsymbol{\theta}^T.
\end{aligned}
$$

$$
\boldsymbol{\theta}^{-1}\hat{\mathbf{P}}\boldsymbol{\theta} = \mathbf{V}\big[\mathbf{V}^T\boldsymbol{\theta}^T\boldsymbol{\theta}\mathbf{V}\big]^{-1}\mathbf{V}^T\boldsymbol{\theta}^T\boldsymbol{\theta}.
$$

Furthermore,

$$
\begin{aligned}
\|\boldsymbol{\theta}^{-1}\hat{\mathbf{P}}\boldsymbol{\theta} - \mathbf{P}\|_2 &= \|\mathbf{V}\big[\mathbf{V}^T\boldsymbol{\theta}^T\boldsymbol{\theta}\mathbf{V}\big]^{-1}\mathbf{V}^T\boldsymbol{\theta}^T\boldsymbol{\theta} - \mathbf{V}\mathbf{V}^T\|_2, \\
&\leq \|\mathbf{V}\big[\mathbf{V}^T\boldsymbol{\theta}^T\boldsymbol{\theta}\mathbf{V}\big]^{-1}\mathbf{V}^T\boldsymbol{\theta}^T\boldsymbol{\theta} - \mathbf{V}\mathbf{V}^T\boldsymbol{\theta}^T\boldsymbol{\theta}\|_2 + \|\mathbf{V}\mathbf{V}^T\boldsymbol{\theta}^T\boldsymbol{\theta} - \mathbf{V}\mathbf{V}^T\|_2, \\
&\leq \|\mathbf{V}\big(\big[\mathbf{V}^T\boldsymbol{\theta}^T\boldsymbol{\theta}\mathbf{V}\big]^{-1} - \mathbf{I}\big)\mathbf{V}^T\|_2 \cdot \|\boldsymbol{\theta}^T\boldsymbol{\theta}\|_2 + \|\boldsymbol{\theta}^T\boldsymbol{\theta} - \mathbf{I}\|_2, \\
&\leq \|\big[\mathbf{V}^T\boldsymbol{\theta}^T\boldsymbol{\theta}\mathbf{V}\big]^{-1}\|_2 \cdot \|\mathbf{I} - \mathbf{V}^T\boldsymbol{\theta}^T\boldsymbol{\theta}\mathbf{V}\|_2 \cdot \|\boldsymbol{\theta}^T\boldsymbol{\theta}\|_2 + \|\boldsymbol{\theta}^T\boldsymbol{\theta} - \mathbf{I}\|_2, \\
&\leq \|\big[\mathbf{V}^T\boldsymbol{\theta}^T\boldsymbol{\theta}\mathbf{V}\big]^{-1}\|_2 \cdot \|\mathbf{I} - \boldsymbol{\theta}^T\boldsymbol{\theta}\|_2 \cdot \|\boldsymbol{\theta}^T\boldsymbol{\theta}\|_2 + \|\boldsymbol{\theta}^T\boldsymbol{\theta} - \mathbf{I}\|_2.
\end{aligned}
$$

We further bound $\|[\mathbf{V}^T\boldsymbol{\theta}^T\boldsymbol{\theta}\mathbf{V}]^{-1}\|_2$.

$$
\begin{aligned}
\|[\mathbf{V}^T\boldsymbol{\theta}^T\boldsymbol{\theta}\mathbf{V}]^{-1}\|_2 &= \left(\lambda_{min}(\mathbf{V}^T\boldsymbol{\theta}^T\boldsymbol{\theta}\mathbf{V})\right)^{-1}, \\
&= \left(\inf_{\|\mathbf{x}\|_2=1}\mathbf{x}^T\mathbf{V}^T\boldsymbol{\theta}^T\boldsymbol{\theta}\mathbf{V}\mathbf{x}\right)^{-1}, \\
&\leq \left(\inf_{\|\mathbf{x}'\|_2=1}(\mathbf{x}')^T\boldsymbol{\theta}^T\boldsymbol{\theta}\mathbf{x}'\right)^{-1}, \\
&= \left(\lambda_{min}(\boldsymbol{\theta}^T\boldsymbol{\theta})\right)^{-1}, \\
&= \|(\boldsymbol{\theta}^T\boldsymbol{\theta})^{-1}\|_2.
\end{aligned}
$$

Hence, we have

$$
\begin{aligned}
\|\boldsymbol{\theta}^{-1}\hat{\mathbf{P}}\boldsymbol{\theta} - \mathbf{P}\|_2 &\leq \left(1 + \|\boldsymbol{\theta}^T\boldsymbol{\theta}\|_2 \cdot \|(\boldsymbol{\theta}^T\boldsymbol{\theta})^{-1}\|_2\right) \cdot \|\mathbf{I} - \boldsymbol{\theta}^T\boldsymbol{\theta}\|_2, \\
&= \left(1 + \kappa(\boldsymbol{\theta})^2\right) \cdot \|\mathbf{I} - \boldsymbol{\theta}^T\boldsymbol{\theta}\|_2.
\end{aligned}
$$

$\square$

**Corollary 1.** *Let $t \geq 1$. Then for each $s = 0, 1, \cdots, t$, we have*

$$
\|\mathbf{P}_s^s - \mathbf{P}_0\|_2 \leq \left(1 + \kappa(\overline{\boldsymbol{\theta}}_s)^2\right) \cdot \|\mathbf{I} - \overline{\boldsymbol{\theta}}_s^T\overline{\boldsymbol{\theta}}_s\|_2,
$$

*where*

- $\overline{\theta} := \boldsymbol{\theta}_{s-1}\cdots\boldsymbol{\theta}_0,\ s \geq 1,$

- $\overline{\theta} := \mathbf{I},\ s = 0.$

Observe that $\mathbf{P}_s^s = (\overline{\boldsymbol{\theta}}_s)^{-1}\mathbf{P}_s\overline{\boldsymbol{\theta}}_s$. Using Lemma 6, we arrive at the main theorem.

**Theorem 1.** *For $t \geq 1$, we have the error estimation*

$$
\|\overline{\mathbf{x}}_{\epsilon,t} - \mathbf{x}_t\|_2^2 \leq \|\boldsymbol{\theta}_{t-1}\cdots\boldsymbol{\theta}_0\|_2^2 \cdot \left(\alpha^{2t}\|\mathbf{z}^\perp\|_2^2 + \|\mathbf{z}^\|\|_2^2 + \gamma_t\|\mathbf{z}\|_2^2\left(\gamma_t\alpha^2(1-\alpha^{t-1})^2 + 2(\alpha - \alpha^t)\right)\right).
$$

*where $\gamma_t := \max_{s\leq t}\left(1 + \kappa(\overline{\boldsymbol{\theta}}_s)^2\right)\|\mathbf{I} - \overline{\boldsymbol{\theta}}_s^T\overline{\boldsymbol{\theta}}_s\|_2$, and $\alpha = \frac{c}{1+c}$, $c$ represents the control regularization. In particular, the equality*

$$
\|\overline{\mathbf{x}}_{\epsilon,t} - \mathbf{x}_t\|_2^2 = \alpha^{2t}\|\mathbf{z}^\perp\|_2^2 + \|\mathbf{z}^\|\|_2^2.
$$

*holds when all $\boldsymbol{\theta}_t$ are orthogonal.*

*Proof.* The input perturbation $\mathbf{z} = \overline{\mathbf{x}}_{\epsilon,0} - \mathbf{x}_0$ can be written as $\mathbf{z} = \mathbf{z}^\| + \cdot\mathbf{z}^\perp$, where $\mathbf{z}^\| \in Z_\|$ and $\mathbf{z}^\perp \in Z_\perp$, where $\mathbf{z}^\|$ and $\mathbf{z}^\perp$ are vectors such that

- $\mathbf{z}^\| \cdot \mathbf{z}^\perp = 0$ almost surely.

- $\mathbf{z}^\|, \mathbf{z}^\perp$ have uncorrelated components.

- $\mathbf{z}^\| \in \mathcal{D}$, and $\mathbf{z}^\perp \in \mathcal{D}^\perp$.

Since $\mathbf{z}^\|$ and $\mathbf{z}^\perp$ are orthogonal almost surely, recall Lemma 4,

$$
\begin{aligned}
\|\overline{\mathbf{x}}_{\epsilon,t} - \mathbf{x}_t\|_2^2 &= \|(\boldsymbol{\theta}_{t-1}\boldsymbol{\theta}_{t-2}\cdots\boldsymbol{\theta}_0)(\mathbf{G}_{t-1}^{t-1}\cdots\mathbf{G}_0^0)z\|_2^2, \\
&\leq \|\boldsymbol{\theta}_{t-1}\boldsymbol{\theta}_{t-2}\cdots\boldsymbol{\theta}_0\|_2^2 \cdot \|(\mathbf{G}_{t-1}^{t-1}\cdots\mathbf{G}_0^0)z\|_2^2, \quad (18)
\end{aligned}
$$

For the term $\|(\mathbf{G}_{t-1}^{t-1}\cdots\mathbf{G}_0^0)z\|_2^2$, recall Lemma 5,

$$
\begin{aligned}
\|(\mathbf{G}_{t-1}^{t-1}\cdots\mathbf{G}_0^0)z\|_2^2 &= \|\left(\alpha^t \cdot \mathbf{I} + (1-\alpha)\sum_{s=0}^{t-1}\alpha^s \cdot \mathbf{P}_s^s\right)\mathbf{z}\|_2^2, \\
&= \|\alpha^t\mathbf{z} + (1-\alpha)\sum_{s=0}^{t-1}\alpha^s\mathbf{P}_0\mathbf{z} + (1-\alpha)\sum_{s=0}^{t-1}\alpha^s(\mathbf{P}_s^s - \mathbf{P}_0)\mathbf{z}\|_2^2, \\
&= \|\alpha^t\mathbf{z} + (1-\alpha^t)\mathbf{z}^\| + (1-\alpha)\sum_{s=0}^{t-1}\alpha^s(\mathbf{P}_s^s - \mathbf{P}_0)\mathbf{z}\|_2^2,
\end{aligned}
$$

in the above, $\mathbf{P}_0$ is an orthogonal projection on $t = 0$ (input data space), therefore, $\mathbf{P}_0 z = \mathbf{z}^\|$. Furthermore, when $s = 0$, $\mathbf{P}_s^s - \mathbf{P}_0 = \mathbf{0}$. Thus,

$$\|(\mathbf{G}_{t-1}^{t-1} \cdots \mathbf{G}_0^0)z\|_2^2$$

$$= \alpha^{2t}\|\mathbf{z}\|_2^2 + (1 - \alpha^t)^2\|\mathbf{z}^\|\|_2^2 + (1 - \alpha)^2 \sum_{s,q=1}^{t-1} \alpha^s \alpha^q \mathbf{z}^T (\mathbf{P}_s^s - \mathbf{P}_0)^T (\mathbf{P}_q^q - \mathbf{P}_0)\mathbf{z}$$

$$+ 2\alpha^t(1 - \alpha^t)\|\mathbf{z}^\|\|_2^2 + 2\alpha^t(1 - \alpha) \sum_{s=1}^{t-1} \alpha^s \mathbf{z}^T (\mathbf{P}_s^s - \mathbf{P}_0)\mathbf{z}$$

$$+ 2(1 - \alpha^t)(1 - \alpha) \sum_{s=1}^{t-1} \alpha^s (\mathbf{z}^\|)^T (\mathbf{P}_s^s - \mathbf{P}_0)\mathbf{z},$$

$$= \alpha^{2t}\|\mathbf{z}^\perp\|_2^2 + \left(\alpha^{2t} + 2\alpha^t(1 - \alpha^t) + (1 - \alpha^t)^2\right)\|\mathbf{z}^\|\|_2^2$$

$$+ (1 - \alpha)^2 \sum_{s,q=1}^{t-1} \alpha^s \alpha^q \mathbf{z}^T (\mathbf{P}_s^s - \mathbf{P}_0)^T (\mathbf{P}_q^q - \mathbf{P}_0)\mathbf{z} + 2\alpha^t(1 - \alpha) \sum_{s=1}^{t-1} \alpha^s \mathbf{z}^T (\mathbf{P}_s^s - \mathbf{P}_0)\mathbf{z}$$

$$+ 2(1 - \alpha^t)(1 - \alpha) \sum_{s=1}^{t-1} \alpha^s (\mathbf{z}^\|)^T (\mathbf{P}_s^s - \mathbf{P}_0)\mathbf{z},$$

$$= \alpha^{2t}\|\mathbf{z}^\perp\|_2^2 + \|\mathbf{z}^\|\|_2^2 + (1 - \alpha)^2 \sum_{s,q=1}^{t-1} \alpha^s \alpha^q \mathbf{z}^T (\mathbf{P}_s^s - \mathbf{P}_0)^T (\mathbf{P}_q^q - \mathbf{P}_0)\mathbf{z}$$

$$+ 2\alpha^t(1 - \alpha) \sum_{s=1}^{t-1} \alpha^s \mathbf{z}^T (\mathbf{P}_s^s - \mathbf{P}_0)\mathbf{z} + 2(1 - \alpha^t)(1 - \alpha) \sum_{s=1}^{t-1} \alpha^s (\mathbf{z}^\|)^T (\mathbf{P}_s^s - \mathbf{P}_0)\mathbf{z}.$$

Using Corollary 1, we have

- 
$$\mathbf{z}^T (\mathbf{P}_s^s - \mathbf{P}_0)\mathbf{z} \le \|\mathbf{z}\|_2^2 \cdot \|\mathbf{P}_s^s - \mathbf{P}_0\|_2,$$
$$\le \gamma_t \|\mathbf{z}\|_2^2.$$

- 
$$\mathbf{z}^T (\mathbf{P}_s^s - \mathbf{P}_0)^T (\mathbf{P}_q^q - \mathbf{P}_0)\mathbf{z} \le \|\mathbf{z}\|_2^2 \cdot \|\mathbf{P}_s^s - \mathbf{P}_0\|_2 \cdot \|\mathbf{P}_q^q - \mathbf{P}_0\|_2,$$
$$\le \gamma_t^2 \|\mathbf{z}\|_2^2.$$

- 
$$(\mathbf{z}^\|)^T (\mathbf{P}_s^s - \mathbf{P}_0)\mathbf{z} \le \gamma_t \|\mathbf{z}^\|\|_2 \cdot \|\mathbf{z}\|_2,$$
$$\le \gamma_t \|\mathbf{z}\|_2^2.$$

Thus, we have

$$\|(\mathbf{G}_{t-1}^{t-1} \cdots \mathbf{G}_0^0)z\|_2^2 \le \alpha^{2t}\|\mathbf{z}^\perp\|_2^2 + \|\mathbf{z}^\|\|_2^2 + \alpha^2(1 - \alpha^{t-1})^2 \gamma_t^2 \|\mathbf{z}\|_2^2 + 2\alpha^{t+1}(1 - \alpha^{t-1})\gamma_t \|\mathbf{z}\|_2^2$$
$$+ 2\alpha(1 - \alpha^t)(1 - \alpha^{t-1})\gamma_t \|\mathbf{z}\|_2^2,$$
$$= \alpha^{2t}\|\mathbf{z}^\perp\|_2^2 + \|\mathbf{z}^\|\|_2^2 + \gamma_t \|\mathbf{z}\|_2^2 \left(\gamma_t \alpha^2(1 - \alpha^{t-1})^2 + 2(\alpha - \alpha^t)\right).$$

Recall the error estimation in Eq. (18),

$$\|\bar{\mathbf{x}}_{\epsilon,t} - \mathbf{x}_t\|_2^2 \le \|\boldsymbol{\theta}_{t-1}\boldsymbol{\theta}_{t-2} \cdots \boldsymbol{\theta}_0\|_2^2 \cdot \|(\mathbf{G}_{t-1}^{t-1} \cdots \mathbf{G}_0^0)z\|_2^2,$$

$$\le \|\boldsymbol{\theta}_{t-1} \cdots \boldsymbol{\theta}_0\|_2^2 \cdot \left(\alpha^{2t}\|\mathbf{z}^\perp\|_2^2 + \|\mathbf{z}^\|\|_2^2 + \gamma_t \|\mathbf{z}\|_2^2 \left(\gamma_t \alpha^2(1 - \alpha^{t-1})^2 + 2(\alpha - \alpha^t)\right)\right).$$

Table 4: ResNet for Both CIFAR-10 and CIFAR-100

| Structure | Configuration |
|---|---|
| Initial Layer | Conv2d (input channel = 3, output channel = 16, kernel size = $3 \times 3$), BatchNorm2d(channel = 16), Relu() |
| Residual Block 1 | {Conv2d (input channel = 16, output channel = 16, kernel size = $3 \times 3$), BatchNorm2d(channel = 16), Relu(), Shortcut()} $\times 6$ |
| Residual Block 2 | Conv2d (input channel = 16, output channel = 32, kernel size = $3 \times 3$), BatchNorm2d(channel = 16), Relu(), Shortcut(), {Conv2d (input channel = 32, output channel = 32, kernel size = $3 \times 3$), BatchNorm2d(channel = 32), Relu(), Shortcut()} $\times 5$ |
| Residual Block 3 | Conv2d (input channel = 32, output channel = 64, kernel size = $3 \times 3$), BatchNorm2d(channel = 64), Relu(), Shortcut(), {Conv2d (input channel = 64, output channel = 64, kernel size = $3 \times 3$), BatchNorm2d(channel = 64), Relu(), Shortcut()} $\times 5$ |
| Final Layer | Fully Connected $(64, 10)$ |

In the specific case, when all $\boldsymbol{\theta}_t$ are orthogonal,

$$\gamma_t := \max_{s \leq t} \left(1 + \kappa(\overline{\boldsymbol{\theta}}_s)^2\right) \|\mathbf{I} - \overline{\boldsymbol{\theta}}_s^T \overline{\boldsymbol{\theta}}_s\|_2$$
$$= 0.$$

Thus,

$$\|\overline{\mathbf{x}}_{\epsilon,t} - \mathbf{x}_t\|_2^2 = \alpha^{2t}\|\mathbf{z}^\perp\|_2^2 + \|\mathbf{z}^\|\|_2^2.$$

$\square$

# B  APPENDIX B  DETAILS OF EXPERIMENTAL SETTING

## B.1  NETWORK CONFIGURATIONS

Since the proposed CLC-NN optimizes the entire state trajectory, it is important to have a relatively smooth state trajectory, in which case, when the reconstruction loss $\|\mathcal{E}_t(\mathbf{x}_t) - \mathbf{x}_t\|_2^2$ at layer $t$ is small, the reconstruction losses at its adjacent layers should be small. For this reason, we use residual neural network (He et al., 2016) as network candidate to retain smoother dynamics. The configuration of the residual neural network used for both CIFAR-10 and CIFAR-100 is shown in Tab. 4.

Based on the configuration of residual neural network shown in Tab. 4, we construct 4 embedding functions applied at input space, outputs of initial layer, residual block 1 and residual block 2. The output of residual block 3 is embedded with a linear orthogonal projection. We randomly select 5000 clean training data to collect state trajectories at all 5 locations.

- For the linear orthogonal projections: we apply the Principle Component Analysis on each of the state collections. We retain the first $r$ columns of the resulted basis, such that $r = \arg\min\{i : \frac{\lambda_1 + \dots + \lambda_i}{\lambda_1 \dots + \lambda_d} \geq 1 - \delta\}$, where $\delta = 0.1$.

Table 5: Convolutional Auto-Encoders

| Structure | Configuration | | | |
|---|---|---|---|---|
| Encoder | Conv2d (input channel = $c_1$, output channel = $c_2$, kernel size = $4 \times 4$, stride = $2 \times 2$, padding = $1 \times 1$ ), ELU(alpha=1), BatchNorm2d(channel = $c_2$), Conv2d (input channel = $c_2$, output channel = $c_3$, kernel size = $4 \times 4$, stride = $2 \times 2$, padding = $1 \times 1$ ), ELU(alpha=1). | | | |
| Decoder | ConvTranspose2d (input channel = $c_3$, output channel = $c_2$, kernel size = $4 \times 4$, stride = $2 \times 2$, padding = $1 \times 1$), ELU(alpha=1), ConvTranspose2d (input channel = $c_2$, output channel = $c_1$, kernel size = $4 \times 4$, stride = $2 \times 2$, padding = $1 \times 1$), | | | |
| Auto-encoder Index | 0 | 1 | 2 | 3 |
| Channel Dimensions $[c_1, c_2, c_3]$ | $[3, 18, 36]$ | $[16, 36, 72]$ | $[16, 36, 72]$ | $[32, 128, 256]$ |

- For the nonlinear embedding: we train $4$ convolutional auto-encoders for the input space, outputs of the initial layer and residual blocks $1, 2$. All of the embedding functions are trained individually. We adopt shallow convolutional auto-encoder structure to gain fast inference speed, in which case, CLC-NN equipped with linear embedding often outperform the nonlinear embedding as shown in Tab. 1. The configuration of all $4$ convolutional auto-encoders are shown in Tab. 5.

### B.2 PERTURBATIONS AND DEFENSIVE TRAINING

In this section, we show details about the perturbations and robust networks that have been considered in this work. For the adversarial training objective function,

$$\min_{\boldsymbol{\theta} \in \Theta} \max_{\mathbf{x}_{\epsilon,0} = \Delta(\mathbf{x}_0, \epsilon)} \mathbb{E}_{(\mathbf{x}_0, \mathbf{y}) \sim \mathcal{D}} [(1 - \lambda) \cdot \Phi_i(\mathbf{x}_{\epsilon,T}, \mathbf{y}, \boldsymbol{\theta}) + \lambda \cdot \Phi_i(\mathbf{x}_T, \mathbf{y}, \boldsymbol{\theta})],$$

where $\Delta(\mathbf{x}_0, \epsilon)$ generates a perturbed data from given input $\mathbf{x}_0$ within the range of $\epsilon$. $\lambda$ balances between standard accuracy and robustness. We choose $\lambda = 0.5$ in all adversarial training.

**For robust networks,** we consider both perturbation agnostic and non-agnostic methods. For the perturbation agnostic adversarial training algorithms equipped $\Delta(\mathbf{x}_0, \epsilon)$, the resulted network that is the most robust against the $\Delta(\mathbf{x}_0, \epsilon)$ perturbation. On the contrary, perturbation non-agnostic robust training methods are often robust against many types of perturbations.

- Adversarial training with the fast gradient sign method (FGSM) (Goodfellow et al., 2014) considers perturbed data as follows.

$$\mathbf{x}_{\epsilon,0} = \mathbf{x}_0 + sign(\nabla_{\mathbf{x}_0} \Phi(\mathbf{x}_T, \mathbf{y})), \quad (\mathbf{x}_0, y) \sim \mathcal{D},$$

where $sign(\cdot)$ outputs the sign of the input. In which case, FGSM considers the worse case within the range of $\epsilon$ along the gradient $\nabla_{\mathbf{x}_0} \Phi(\mathbf{x}_T, \mathbf{y})$ increasing direction. Due to the worse case consideration, it does not scale well for deep networks, for this reason, we adversarially train the network with FGSM with $\epsilon = 4$, which is half of the maximum perturbation considered in this paper.

- The label smoothing training (Label Smooth) (Hazan et al., 2017) does not utilize any perturbation information $\Delta(\mathbf{x}_0, \epsilon)$. It converts one-hot labels into soft targets by setting the correct class as $1 - \epsilon$, while other classes have value of $\frac{\epsilon}{N-1}$, where $\epsilon$ is a small constant and $N$ is number of classes. Specifically, we choose $\epsilon = 0.9$ in this paper.

Table 6: Experimental results on DenseNet-40 from standard training.

| Dataset | $\epsilon$ | Accuracy: original without CLC / CLC-NN + Nonlinear Embedding | | | | |
| --- | --- | --- | --- | --- | --- | --- |
| | | Type of input perturbations | | | | |
| | | None | Manifold | FGSM | PGD | CW |
| CIFAR-10 | 2 | | 19 / 67 | 27 / 57 | 0 / 52 | 5 / 79 |
| | 4 | 92 / 89 | 4 / 56 | 20 / 48 | 0 / 37 | 0 / 79 |
| | 8 | | 1 / 59 | 15 / 33 | 0 / 17 | 0 / 79 |
| CIFAR-100 | 2 | | 8 / 25 | 8 / 24 | 0 / 25 | 3 / 47 |
| | 4 | 70 / 64 | 2 / 24 | 4 / 15 | 0 / 10 | 0 / 45 |
| | 8 | | 1 / 24 | 3 / 7 | 0 / 5 | 0 / 45 |

- Adversarial training with the project gradient descent (PGD) (Madry et al., 2017) generates adversarial data by iteratively run FGSM with small step size, which results in stronger perturbations compared with FGSM within the same range $\epsilon$. We use 7-step of $\epsilon = 2$ to generate adversarial data for robust training.

**For Perturbations,** we consider the maximum range of $\epsilon = 2, 4, 8$ to test the performance the network robustness against both strong and weak perturbations. For this work, we test network robustness with the manifold-based attack (Jalal et al., 2017), FGSM (Goodfellow et al., 2014), 20-step of PGD (Madry et al., 2017) and the CW attack (Carlini & Wagner, 2017).

### B.3 ONLINE OPTIMIZATION

**Optimization Methods.** we use Adam (Kingma & Ba, 2014) to maximize the Hamiltonian Eq. (9) with default setting. In which case, solving the PMP brings in extra computational cost for inference.

Each online iteration of solving the PMP requires a combination of forward propagation (Eq. (7)), backward propagation (Eq. (8)) and a maximization w.r.t. the control parameters (Eq. (9)), which has computational cost approximately the same as performing gradient descent on training a neural network for one iteration. For the numerical results presented in the paper, we choose the maximum iteration that gives the best performance from one of $[5, 10, 20, 30, 50]$.

## C MORE NUMERICAL EXPERIMENTS

The proposed CLC-NN is designed to be compatible with existing open-loop trained. We show extra experiments by employing the proposed CLC-NN on two baseline models, DenseNet-40 (Table 6).

The layer-wise projection performs orthogonal projection on the hidden state. We define the local cost function at the $t^{th}$ layer as follows

$$J(\mathbf{x}_t, \mathbf{u}_t) = \frac{1}{2}\|\mathbf{Q}_t(\mathbf{x}_t + \mathbf{u}_t)\|_2^2 + \frac{c}{2}\|\mathbf{u}_t\|_2^2,$$

the layer-wise achieves the optimal solution at local time $t$,

$$\mathbf{u}_t^*(\mathbf{x}_t) = \arg\min_{\mathbf{u}_t} J(\mathbf{x}_t, \mathbf{u}_t).$$

However, the layer-wise optimal control solution does not guarantee the optimum across all layers. In Table 7, we launch comparisons between the proposed CLC-NN with layer-wise projection. Specifically, under all perturbations the proposed CLC-NN outperforms layer-wise projection.

## D ROBUSTNESS AGAINST MANIFOLD-BASED ATTACK

The manifold-based attack (Jalal et al., 2017) (denoted as Manifold) has shown great success on breaking down the manifold-based defenses (Samangouei et al., 2018). The proposed CLC-NN can successfully defend such specifically design adversarial attack for manifold-based defenses and improves the robustness accuracy from $1\%$ to $81\%$ for the standard trained model in Cifar-10, and $2\%$ to $52\%$ in Cifar-100.

Table 7: Comparison between CLC-NN and layer-wise projection.

| Dataset | $\epsilon$ | Accuracy: Layer-wise projection / CLC-NN + Linear embedding | | | | |
|---------|-----------|------|----------|------|------|------|
| | | | Type of input perturbations | | | |
| | | None | Manifold | FGSM | PGD | CW |
| CIFAR-10 | 2 | 92 / 88 | 68 / 79 | 29 / 56 | 4 / 50 | 51 / 75 |
| | 4 | | 64 / 78 | 15 / 40 | 0 / 31 | 50 / 75 |
| | 8 | | 64 / 78 | 10 / 20 | 0 / 11 | 49 / 76 |
| CIFAR-100 | 2 | 67 / 60 | 45 / 51 | 13 / 25 | 2 / 17 | 35 / 47 |
| | 4 | | 44 / 50 | 8 / 15 | 0 / 6 | 34 / 47 |
| | 8 | | 44 / 50 | 5 / 9 | 0 / 1 | 34 / 47 |

We provide detailed explanation for the successful defense of the proposed CLC-NN against such strong adversarial attack. Exsiting manifold-based defense (Samangouei et al., 2018) focuses on detecting and de-noising the input components that do not lie within the underlying manifold. The overpowered attack proposed in Jalal et al. (2017) searches adversarial attack with in the embedded latent space, which is undetectable for the manifold-based defenses and caused complete failure defense.

In the real implementation, the manifold-based attack (Jalal et al., 2017) is detectable and control-lable under the proposed framework due to the following reason. **The numerically generated man-ifold embedding functions are not ideal.** The error sources of non-ideal embedding functions are mainly due to the algorithm that used to compute the manifold, the architecture of embedding func-tion, and the distribution shift between training and testing data (embedding functions of training data do not perfectly agree with testing data). In which case, even the perturbation is undetectable and non-controllable at initial layer, as it is propagated into hidden layers, each layer amplifies such perturbation, therefore, the perturbation becomes detectable and controllable in hidden layers.

We randomly select the batch of testing data to generate the manifold-based attack following the same procedure proposed in Jalal et al. (2017). The proposed method improves the attacked accuracy from $1\%$ to $78\%$. More specifically, we compare the differences of all hidden states spanning the orthogonal complement between a perturbed testing data and its unperturbed counterpart, $\|\mathbf{P}_t^\perp \mathbf{x}_{\epsilon,t} - \mathbf{P}_t^\perp \overline{\mathbf{x}}_{\epsilon,t}\|$, where $\mathbf{P}_t^\perp$ is a projection onto the orthogonal complement. The difference is growing such as $0, 0.016, 0.0438, 0.0107, 0.0552$ for hidden states at layer $0, 1, 2, 3, 4$ respectively. This validates the argument for how the proposed method is able to detect such perturbation and controls the perturbation in hidden layers.

Furthermore, we provide some insights about the reasons behind the success of such an adversarial attack. This follows the same concept of the existence of adversarial attack in neural networks. The highly nonlinear behaviour of neural networks preserves complex representative ability, meanwhile, its powerful representation results in its vulnerability. For example, a constant function has $50\%$ chance to make a correct prediction in binary classification problem under any perturbation, but its performance is limited. Therefore, we propose to use a linear embedding function that compensates between the embedding accuracy and robustness.

## E  DEFINITION OF THREAT MODEL

Generally, an attacker should not have access to the hidden states during inference, in which case, an attacker is not allowed to inject extra noise during inference. To define the threat model of the proposed method, for the white-box setting, an attacker has access to both network and *all* embedding functions. The condition that the perturbation $\epsilon \cdot \mathbf{z}$ makes our method vulnerable is defined as follows,

$$\sum_{t=0}^{T-1} \|\mathcal{E}_t(\mathbf{x}_{\epsilon,t}) - \mathbf{x}_{\epsilon,t}\|_2^2 = 0, \quad \mathbf{x}_{\epsilon,0} = \mathbf{x}_0 + \epsilon \cdot \mathbf{z}.$$

In words, the perturbation $\epsilon \cdot \mathbf{z}$ applied on the input data must result in $0$ reconstruction losses across all hidden layers, which means its corresponding state trajectory does not span any of all orthogonal complements of all hidden state spaces. To obtain an effective attack satisfying the above equation, conventional gradient-based attackers cannot guarantee to find an perfect attack. A possible way is

to perform grid-search backward in layers to find such an adversarial attack satisfying the thread model condition, which is extremely costly.

