# OpenReview forum: "Towards Robust Neural Networks via Close-loop Control"
_ICLR.cc/2021/Conference — ICLR 2021 Poster_

### Official Review · AnonReviewer3 · 2020-10-28
**This paper argues to proposed a closed-loop control in the robustness training to achieve good results on different types of attack. To me, this statement is ambitious to be called closed-loop control, but the overall structure is meaningful and interesting.**

**Rating:** 7
**Confidence:** 4

**Review:**

Strength:
1. This paper first introduced layer-wised projection from the poisoned data to the clean data.
2. The results show improvement of the robustness over the baseline on different types of attacks.

Weakness:
1. The statement of the closed-loop control is a little bit ambitious. The overall methods are the layer-wise projection from the poisoned data to the clean data manifold. Normally, in the closed-loop, we will use the control signal $u$ to control the original data instead of the next layer data. For example, the final balance stage should be $u=g(x+f(x+u))=0$. So closed-loop should be at least multi-steps within one layer. For different layers, the closed-loop control will have different control signals, since the dimension/ distribution between layers is much different. So this method is only a one-step layer-wise projection. The $x$ between layers cannot be viewed as the same sample to be controlled. I would recommend the author to change the statement from closed-loop into the layer-wise projection for a better suit. Also, this method is still a feed-forward network, not a "loop" control. I do think this method is interested, just a little ambitious. This could be a useful extension of the resnet-based network, since the control $u$ can be viewed as a complicated version of residuals.
2. The experiment is weak for only comparing with one baseline. Also, can the author provide which baseline model that the author is comparing with? I cannot find it in the text. I would appreciate it.
3. It's unclear to me about the training of $\mathcal{E}(x)$, will this requires extra data to train? What's the running speed of this "closed-loop" method compared with others?

Some tiny comments:
1. A trivial comparison would be to train an autoencoder for each layer of $x$ and only use the decoded results to pass through the network. This in principle learns the data manifold and provides the projection to this manifold.
2. Table 2, the dataset name is not aligned in the center.
3. Table 3 should be more self-explainable. It's a little confusing in the current form.
4. Have the author tried multi-steps in a single layer or constraint $x_t$ to be in the same space?



----- post rebuttal -----

The authors addressed most of my concerns and the revision is better than before.
I would like to increase my score and would recommend an acceptance.

---

> ### Author Response · Authors · 2020-11-18
> **We have addressed all of your questions and provided explanation about the close-loop control.**
>
> Thank you for your opinions, we provide detailed explanations to your concerns.
>
> (1).**The statement of the closed-loop control is a little bit ambitious**.  Our method is indeed a close-loop feedback control. Please note that $\mathbf{x}_0$ (the data input of a neural network) should NOT be regarded as the input signal of a dynamic system. Instead, $\mathbf{x}_0$ is just an initial condition, and the excitation input signal is $\mathbf{u}_t$ (which is 0 in our case when there is no control). Therefore, the forward map is from $\mathbf{u}_t$ to internal hidden states $\mathbf{x}_t$ and to output label $\mathbf{y}$.  **The mapping from internal state $\mathbf{x}_t$ to ${\cal E}_t$ and to $\mathbf{u}_t$ in our paper forms a feedback path and closes the whole loop.**
>
> (2).**Can the author provide which baseline model that the author is comparing with**. We have compared against __5 specific baseline models__ (standard training in Table 1, three open-loop robust training models in Table 2, and reactive defense in Table 3) against a different attacks (manifold, FGSM, CW, PGD). In this paper, we conduct experiments on ResNet-20, the details of which is provided in Appendix B. In appendix C, we have added an extra set of experiments on DenseNet-40.
>
> Specifically, in Table 1 and Table 2, since our proposed close-loop control model is designed to be compatible with pre-trained networks, we compared with various open-loop trained models by equipping the close-loop framework.
>
> (3).**It’s unclear to me about the training of $\mathcal{E}_t$, will this requires extra data to train? What’s the running speed of this ”closed-loop” method compared with others?** The training of $\mathcal{E}_t$ requires neither extra data, nor any attack information. The proposed method is truly attack-agnostic. Specifically, we only choose $6000$ training data randomly for linear embedding functions and the entire training data set for nonlinear embedding functions.
>
> The running speed depends on the online iterative version of solving the maximization condition. Each online iteration of the PMP solver requires a forward propagation (Eq. 7), a backward propagation (Eq. 8) and a maximization step (Eq. 9), which has a similar cost of one gradient-descent step in the training process. In this paper, we choose the maximum number of iterations from [5, 10, 20, 30, 50] that gives the best performance.
>
> (4).**A trivial comparison would be to train an autoencoder for each layer of x and only use the decoded results to pass through the network**. Here, we present the comparison between our method with the above suggested method. In Table 7 from the updated script, the proposed method outperforms the layer-wise projection that is suggested by the reviewer against all perturbations.
>
> (5).**Have the author tried multi-steps in a single layer or constraint $x_t$ to be in the same space?** We have considered this setting. Multi-steps in single layer implies the embedding functions to be recurrent neural networks to utilize the history information. This can greatly increase the online inference overload, therefore, we choose a single step scheme at each layer.
>
> Constraining $\mathbf{x}_t$ to be in the same space implies applications with recurrent neural networks, in which case, all hidden state and input are in the same space and can be controlled by a single controller. We leave this for future investigation.
>
> __We have also fixed all of your minor comments in the updated manuscript.__

---

### Official Review · AnonReviewer1 · 2020-10-28
**Interesting approach ..**

**Rating:** 6
**Confidence:** 4

**Review:**

The paper builds on recent revival in control-theoretic approaches to deep neural networks by proposing an adaptive controller that projects intermediate representations in the network to their "manifolds" and consequently makes the neural network robust to input perturbations.

Pros:

+ Active controller-based projection of intermediate features is an interesting idea and the utility of Pontryagin's maximum principle to address the challenge of high dimensionality of the state (features) is a good observation.

Cons:

- The manifold-based defense has been shown to be broken previously. For e.g. please see section 4 of https://arxiv.org/pdf/1712.09196.pdf . Manifold/GAN/VAE based defenses can be easily broken by just attacking the projection network and the original network together. The paper considers manifold based defense at all layers using an active controller doing the projection. While comparison with pixeldefense is useful, it would be good to launch an attack similar to the reference above and then observe the effectiveness of the approach. The arguments in the paper are not sufficient to convince the reviewer that this defense is practical. The paper is severely lacking in comparison with existing defense approaches. The state of the art for the used dataset in the paper is significantly better than the effectiveness of the presented approach. For e.g. see https://github.com/MadryLab/robustness

In summary, the paper is a good effort to exploit the use of active controllers in deep learning. But firstly, the use of manifold-based projection as a cost function is itself a non-robust defense against adversarial examples. Second, the experimental evaluation in the paper is significantly lacking and does not meet the standards of a venue such as ICLR. The reviewer will strongly recommend reviewing the advices in https://arxiv.org/abs/1902.06705 on this topic. At this point, the paper is interesting but it needs significant development and is not yet ready for publication.

Questions for the author:

1. How critical is Pontryagin’s Maximum Principle to the presented approach? What prevents one from using projection to a lower dimensional embedding space followed by a state space control method? In particular, if one is building on the manifold assumption, then isn't it reasonable to not worry about high dimensionality for designing the controller too?

2. Is it realistic to assume the "input perturbation to be a random vector" in Section 5 for theoretical analysis when we are considering adversarial attacks such as PGD, CW? If not, then isn't theorem 1 not relevant to the primary topic of the paper?

------ After author's response:

* The response of authors identifies the problem of using running loss in projected space. While one can try to get around it by projecting the loss function as well but that would be a convoluted way to solve the problem, and in any case, not a strong criticism of the presented approach.

* The updated document has generalized the derivation to take general perturbations into account.

* Updates Tables 1-3 resolve empirical analysis concerns of the reviewer.

With these improvements, the reviewer is happy to recommend acceptance of the paper.

---

> ### Author Response · Authors · 2020-11-18
> **We have addressed all questions (including the one about manifold attacks) and provided detailed explanations.**
>
> We appreciate your comments that have greatly improved the paper.
>
> __1. The manifold-based defense has been shown to be broken previously. ... The arguments in the paper are not sufficient to convince the reviewer.__
>
>  __(1.1) Our method actually can effectively handle the manifold-based attacks__. This is shown in the updated Tables 1-3. Specifically, when a standard trained model is attacked by the manifold-based attack Jalal et al.(2017), our method has improved the robust accuracy from 1% to 81% on CIFAR-10 and from 2% to 52% on CIFAR-100 (see Table 1). Similar improvements have been achieved by our method on the various robustly trained models (see Table 2).
>
>  __(1.2) We provide some explanations here.__  Samangouei et al., 2018 detects and de-noises the input components that are outside the underlying manifold. The attack proposed in Jalal et al.(2017) searches adversarial attack within the embedded latent space, which cannot be detected by manifold-based defenses and thus can cause complete defense failures.  __In practice, the manifold-based attack Jalal et al.(2017) is detectable and controllable by our framework due to two reasons.__ **Firstly, the numerically generated manifold embedding is not ideal.** The error of an non-ideal embedding function is caused by the inexact numerical computation, the architecture of embedding function, the distribution shift between training and testing data (the embedding functions of training and testing data are not exactly the same). **Secondly, the components that are orthogonal to the true manifold can be amplified and detected in forward propagation.**  To see this, we randomly select a batch of testing data to generate the manifold attack. We compare the differences of all hidden states spanning the orthogonal complement between a perturbed testing data and its unperturbed counterpart, || $\mathbf{P}_t \mathbf{x}_\epsilon - \mathbf{P}_t \overline{\mathbf{x}}_\epsilon $||, where $\mathbf{P}_t$ is a projection onto the orthogonal complement. The difference grows from 0 to 0.016, 0.0438, 0.0107, 0.0552 at layer 0, 1, 2, 3, 4 respectively. The difference at the hidden layers are detected by our framework, and the robust accuracy increases from 1% to 78%.
>
> __(1.3). We define our threat model in the white-box setting__, where an attacker has access to both network and embedding functions. Our model becomes vulnerable if, $\sum_{t=0}^{T-1} || \mathcal{E}_t(\mathbf{x}_\epsilon) - \mathbf{x}_\epsilon ||_2^2 = 0$, s.t. $\mathbf{x}_\epsilon = \mathbf{x}_0 + \mathbf{z}$. The perturbation $\mathbf{z}$ must result in $0$ reconstruction losses across all layers, which means its state trajectory stays inside the manifolds of all hidden states. An attacker cannot alter the hidden states or inject perturbations to hidden layers during inference. Gradient-based methods cannot satisfy the above conditions to obtain an end-to-end effective attack. A possible attack is to perform grid-search backward through all layers, which is extremely costly.
>
> More details are provided in Appendix D.
>
> __2. The paper is severely lacking in comparison with existing defense approaches.__ The network used in Madry et al. (2017) is wide ResNet-50. Compared with ResNet-20 in this paper, wide ResNet-50 has more expressive power in adversarial training and results in better baseline performance. In our updated Table 2, the model trained PGD is comparable with the state of the art.  Note that our method is applicable to many networks, so we focus on how to enhance the model robustness under various attacks.
>
> __3. How critical is Pontryagin’s Maximum Principle to the presented approach?__ The PMP is a key enabler to make the computation numerically feasible. We choose the PMP to avoid the curse of dimensionality of solving the optimal control globally, the computational cost from high dimensional system, such as neural networks, is intractable.
>
> __4. What prevents one from using projection to a lower dimensional embedding space followed by a state space control method?__ The running loss (Eq. 3) prevents us from performing a state-space control in a projected lower dimensional embedding space. The loss function relies on measuring the orthogonal complement components. In the embedding sub-space, the running loss is difficult to define.
>
> __5. Is it realistic to assume the "input perturbation to be a random vector"?__ In the updated manuscript, we have generalized our derivation to take general perturbations into account. For **any perturbation** $\mathbf{z}$ (random vector or adversarial attack), s.t. $\mathbf{z} = \mathbf{z}^{\parallel} + \mathbf{z}^{\perp}$, where $\mathbf{z}^{\parallel} \in Z_{\parallel}$ and $\mathbf{z}^{\perp} \in Z_{\perp}$, the error estimation from Theorem $1$ shows the working principle behind the proposed method. Specifically, the perturbation from the orthogonal complement of the input space ($\mathbf{z}^{\perp}$) are controlled by the proposed method.

---

> > ### Comment · AnonReviewer1 · 2020-11-24
> > **Thank you!**
> >
> > The response has clarified the main concerns of the reviewer.

---

### Official Review · AnonReviewer4 · 2020-10-28
**Technically sound study and convincing empirical results**

**Rating:** 7
**Confidence:** 3

**Review:**

### Summary
This study develops a closed-loop control strategy to improve robustness of neural networks to adversarial attacks. The study is technically sound and the empirical results on classification tasks are convincing.


### Quality

The paper is technically sound and the claims are appropriately backed by empirical evaluation. However, I would recommend the authors to discuss a bit more the additional computational cost of running the closed-loop method.


### Clarity

The manuscript is clearly written and provides enough information for an expert reader to understand all the steps to reproduce the results.


### Originality

The novelty of the study resides in the development of a closed-loop control method for increasing the robustness of neural networks. The strategy is devised to scale to the typical high-dimensional nature of neural network activations.


### Significance of the work

The results suggest that the developed approach is a solid step towards developing robust neural networks.


### Some typos:

-instead of "cause different data distribution deviating", "cause data distributions to deviate";

-instead of "The resulting control policy [...] make it", "The resulting control policy [...] makes it";

-instead of "the embedding are effective", "the embeddings are effective";

-instead of "the perturbed states in Fig.2 [...] has", "the perturbed states in Fig.2 [...] have";

-instead of "to obtain all the intermediate hidden states [...] and accumulates", "to obtain all the intermediate hidden states [...] and to accumulate";

-issue with reference "E. 2017".

---

> ### Author Response · Authors · 2020-11-18
> **All questions has been answered. Computational cost of the close-loop control is discussed.**
>
> Thank you for your positive opinions, your feedback indeed has emphasized on of your main contributions.
>
>  (1).Indeed, existing works that intend to connect optimal control theory and neural networks have focused on the training of neural networks (Li et al., 2017; Liu et al., 2020) with open-loop control methods (fixed parameters) (Zhang et al., 2019). We want to emphasize one of the contributions of this paper is to introduce a formulation of running loss (Eq. 3 and Eq. 4) as a promising initial step to connect close-loop control theory with neural networks to inspire new defense algorithms. Under this framework, more dedicated cost function (Eq. 3) can be further investigated to improve performance. Both numerical experiments and theoretical analysis have show the promises for this direction.
>
> (2). __I would recommend the authors to discuss a bit more the additional computational cost of running the closed-loop method__ Solving the PMP brings in some extra computational cost for inference. This computational overhead may be mitigated in the future by solving the high-dimensional Hamilton Jacobi Bellman (HJB) PDE off-line. Currently, solving the resulting high-dim HJB-PDE is still intractable despite the recent progress in numerical solvers  (Han et al., 2018). Therefore, in this paper we chose to solve an online PMP problem. The PMP problem has a much lower computational cost, but it cannot be done offline, thus leads to computational overhead in inference. Specifically, each online iteration of the PMP solver requires a forward propagation (Eq. 7), a backward propagation (Eq. 8) and a maximization step (Eq. 9), which has a similar cost of one gradient-descent step in the training process.
>
> We have summarized this discussion in Appendix B. 3.
>
> References
>
> [1].Jiequn Han, Arnulf Jentzen, and E Weinan.  Solving high-dimensional partial differential equationsusing  deep  learning.Proceedings of the National Academy of Sciences,  115(34):8505–8510,2018.
>
> [2].Qianxiao Li, Long Chen, Cheng Tai, and E Weinan.  Maximum principle based algorithms for deeplearning.The Journal of Machine Learning Research, 18(1):5998–6026, 2017.
>
> [3].Guan-Horng Liu, Tianrong Chen, and Evangelos A Theodorou. Differential dynamic programmingneural optimizer.arXiv preprint arXiv:2002.08809, 2020.
>
> [4].Dinghuai Zhang, Tianyuan Zhang, Yiping Lu, Zhanxing Zhu, and Bin Dong.  You only propagateonce: Accelerating adversarial training via maximal principle. InAdvances in Neural InformationProcessing Systems, pp. 227–238, 2019.

---

### Official Review · AnonReviewer2 · 2020-10-29
**Increasing robustness via optimal control**

**Rating:** 7
**Confidence:** 3

**Review:**

Keeping the performance of deep neural networks against data perturbations is an important and open problem. The authors propose an optimal control-based approach by taking dynamical systems perspective. The proposed method sounds intuitive and efficient. Authors supply theoretical analysis and (small) experimental evaluation. Overall, I believe paper is a good. However, I would like to get some points clarified:
a)	Authors used manifold assumption (which is a reasonable assumption for many problems) to define running loss (eq 3). (If I am not mistaken) They choose a quadratic loss to have a tractable optimization problem. However, under these assumptions, one may choose many different losses. Would you please comment on the form of the loss and its impact to the method?
b)	Let’s assume dynamical systems perspective is a right perspective for analysing deep neural networks (to be honest I don’t have any criticism about this). To use control theoretic tools, one needs to comment on controllability and observability of the controlled system. I suspect these mentioned properties are a function of the neural network architecture or do authors think the proposed method (as shown in Figure 1) makes each and every deep neural network architecture controllable and/or observable? I would like to hear authors perspective on these issues.
c)	As I mentioned before, the empirical study is quite small, and I didn’t see any baseline (do I miss something here). Do authors consider extending their empirical study and compare their method with some baselines.

I would like to emphasize one more time that, I am positive about the paper. However, I would like to note that I am not expert in the field and I am open to change my view in both direction.

---

> ### Author Response · Authors · 2020-11-18
> **Clarifications have been provided and the paper has been updated to address your great questions.**
>
> Thank you for your positive opinion and great feedback. We highly appreciate it.
>
> (1). __Would you please comment on the form of the loss and its impact to the method?__ The loss function (Eq. 3) is used to measure the deviation between hidden state of testing data and the underlying true data manifold, while satisfying a control regularization. In the high-dimensional case, the $L_2$ norm measurement is reasonable and effective to detect such deviation. Quadratic loss is often employed to measure noise (in our case, perturbation), it also well-aligns with the conventional linear quadratic regulator control and enables more intuitive theoretical derivation as shown in Eq. (10, 11). Other loss functions can be further investigated, such as Huber loss, M-estimator.
>
> (2).__Do authors think the proposed method (as shown in Figure 1) makes each and every deep neural network architecture controllable and/or observable?__ Controllability and observability are indeed essential considerations behind our proposed framework. For the proposed framework, $\mathbf{x}_{t+1} = \boldsymbol\theta_t (\mathbf{x}_t + \mathbf{u}_t)$, we use full state feedback information $\mathbf{x}_t$ to generate the control $\mathbf{u}_t$. Meanwhile, the generated control $\mathbf{u}_t$ lies in the same space as the corresponding state space, s.t. $\mathbf{u}_t, \mathbf{x}_t \in \mathbb{R}^d$. The proposed method indeed has full controllability and observability.
>
>    Furthermore, controllability also depends on the underlying system. In general cases, when the system is non-degenerate (e.g. linear dynamics where $\boldsymbol\theta_t$ are all invertible), full controllability and observability are perserved. Neural networks generally preserve such condition, which results in full controllability and observability.
>
>    The reviewer's comment can lead to an interesting direction as future investigations. To reduce both memory cost and inference time, it is worth further investigating how to generate accurate low-dimensional control signals to control the entire dynamic.
>
> (3). __The empirical study is quite small, and I didn’t see any baseline.__ Sorry for the confusion. __We have compared with $5$ baseline models__ in our original manuscript, including an uncontrolled model from standard training (Table 1), three models from robust training (e.g., robust training with FGSM, PGD, and label smoothing in Table 2), and a reactive defense method (Table 3). Furthermore, we have enriched our results by adding some experiments on the DenseNet-$40$ model in Appendix C. In addition, as suggested by the AnonReviewer $1$, in our updated manuscript, we replace the random perturbation by the manifold-based attack in Table 1, Table 2, and Table 3, the results have shown promises.
>
>    Since our proposed close-loop control model is designed to be compatible with pre-trained networks, we compared with various open-loop trained models by equipping the close-loop framework in Table $1$ and Table $2$. The numerical results have shown that our close-loop control can indeed significantly improve the robustness of various models under a broad class of attacks/perturbations.
>
>    When comparing with reactive defenses in Table 3, we have used the same embedding functions for both methods. Note that our method can be equipped with any generative models as used in Samangouei et al. (2018), Song et al. (2017). Due to the limited computational resource, we only present the shallow embedding functions (see Appendix B for details).
>
> References
>
> [1].P. Samangouei, M. Kabkab, and R. Chellappa. Defense-gan: Protecting classifiers againstadversarial attacks using generative models. arXiv:1805.06605, 2018.
>
> [2].Y. Song,  T. Kim,  S. Nowozin,  S. Ermon,  and N. Kushman.   Pixeldefend:Leveraging  generative  models  to  understand  and  defend  against  adversarial  examples. arXiv:1710.10766, 2017.

---

### Author Response · Authors · 2020-11-19
**Details for changes made in the updated version**

We thank comments from all reviewers. We have addressed all comments and updated the paper. Here, we point out the changes made during this revision.

1.In section 5, we have modified our main theorem to be applicable for **any** perturbations. The newly derived upper bound is more general, strictly tighter, and aligns with the previous error estimation for the special case. We also added comments on the new upper bound for the error estimation.

2.We updated our numerical experiments in all Table 1, 2, 3. We use the same choice of hyper-parameter (learning rate and maximum iterations) to solve the PMP for all perturbations, which truly shows that our model is robust against various perturbations without any knowledge of perturbations.

3.In Table 1, 2, 3, we consider the manifold-based attack, which was previously shown that manifold-based defenses are vulnerable against it.

4.In Appendix A, we have added Lemma 6 and modified the derivation of Theorem 1.

5.In Appendix C, we added more experiments for another network architecture, and comparison with layer-wise projection.

6.In Appendix D, we added explanations for reasons behind the success of our proposed method defending the manifold-based attack.

7.We provided the definition of our threat model in Appendix E.

---

### Decision · Program_Chairs · 2021-01-07
**Final Decision**

**Decision:**

Accept (Poster)

**Comment:**

Thank you for your submission to ICLR.  The reviewers and I are in agreement that the work presents some interesting connections between closed-loop control and stabilization of activations to an observed manifold.  Specifically, the idea of using optimal control dynamic programming techniques to compute optimal adjustments to ensure control on this manifold is an interesting one and may have other implications within deep networks.

Although the reviewers were convinced by the experiments on robustness, I remain a bit skeptical here.  The results show that while the method marginally improves robustness to small-epsilon perturbations, the models are still quite non-robust against the size perturbations frequently used in assessing adversarial robustness (e.g., to eps=8 perturbations on CIFAR10, where the best approach gets ~11% accuracy against PGD attacks).  It doesn't really matter how well a defense works against sub-optimal attacks: if PGD is able to decrease its accuracy this much, clearly the model is not very robust (and it seems upon reading that only 20-step PGD, with no restarts, was used as an attack, which is a fairly weak variant of PGD).  Furthermore, the approach didn't improve much upon PGD-based adversarial training when combined with it, either, overall suggesting that the impact on robustness is somewhat minor, and needs to be evaluated quite a bit more.

While I don't believe these concerns are substantive enough to override the beliefs of all reviewers, I think that the authors could do a much better job of evaluating the actual robustness of these models (following the advice of https://arxiv.org/abs/1902.06705).  And if the resulting metrics are not as strong as hoped for, then it would be good to evaluate other possible benefits of the approach (perhaps to random distribution shift? it seems a much more likely situation for there to be real gains?).  Thus, while I believe the paper has some interesting ideas, I think the authors should probably tone down some of the current claims of improving adversarial robustness unless they can provide a much more thorough evaluation.